# Ion currents through Kir potassium channels are gated by anionic lipids

Ruitao Jin[1,9], Sitong He[1,9], Katrina A. Black[2,3,9], Oliver B. Clarke[4], Di Wu [5,6], Jani R. Bolla [6,7], Paul Johnson[8], Agalya Periasamy[2,3], Ahmad Wardak [2,3], Peter Czabotar [2,3], Peter M. Colman [2,3], Carol V. Robinson [5,6], Derek Laver[8], Brian J. Smith [1✉] & Jacqueline M. Gulbis [2,3✉]

Ion currents through potassium channels are gated. Constriction of the ion conduction pathway at the inner helix bundle, the textbook gate of Kir potassium channels, has been shown to be an ineffective permeation control, creating a rift in our understanding of how these channels are gated. Here we present evidence that anionic lipids act as interactive response elements sufficient to gate potassium conduction. We demonstrate the limiting barrier to $K^+$ permeation lies within the ion conduction pathway and show that this gate is operated by the fatty acyl tails of lipids that infiltrate the conduction pathway via fenestrations in the walls of the pore. Acyl tails occupying a surface groove extending from the cytosolic interface to the conduction pathway provide a potential means of relaying cellular signals, mediated by anionic lipid head groups bound at the canonical lipid binding site, to the internal gate.

[1] La Trobe Institute for Molecular Science, La Trobe University, Melbourne, VIC 3086, Australia. [2] Structural Biology Division, The Walter and Eliza Hall Institute of Medical Research, Parkville, VIC 3052, Australia. [3] Department of Medical Biology, The University of Melbourne, Parkville, VIC 3052, Australia. [4] Department of Anesthesiology and Department of Physiology and Cellular Biophysics, Columbia University, New York, NY 10032, USA. [5] Physical and Theoretical Chemistry Laboratory, University of Oxford, Oxford OX1 3QZ, UK. [6] The Kavli Institute for Nanoscience Discovery, Oxford OX1 3QU, UK. [7] Department of Plant Sciences, University of Oxford, Oxford OX1 3QU, UK. [8] School of Biomedical Sciences and Pharmacy, University of Newcastle and Hunter Medical Research Institute, Newcastle, NSW 2308, Australia. [9]These authors contributed equally: Ruitao Jin, Sitong He, Katrina A. Black. ✉email: brian.smith@latrobe.edu.au; jgulbis@wehi.edu.au

**U**ntil recently, it was thought that K$^+$ channels must adopt a wide pore conformation to enable ion permeation[1–3]. However, K$^+$ can be conducted through a constriction at the intracellular entrance[4] only marginally wider than twice its ionic radius of 1.33 Å (Pauling)[5]–1.38 Å[6], due to transient depletion and replenishment of its hydration shell[4]. The capacity of K$^+$ ions to move through a constricted inner helix bundle has re-opened the question of gating, or how Kir channels achieve controlled switching between conducting and non-conducting physiological states.

Eukaryotic Kir channels are regulated by phosphoinositides[7]. A signature of the lipid-binding site is a five-residue basic sequence on the intracellular extension of the inner helices, typically starting with Arg-Pro (Supplementary Fig. 1). Kir channels are unresponsive until a phosphoinositide (PI) head group binds, after which they gate and conduct normally[8,9]. While the binding site can accommodate other lipids[10] and detergents[11], only PI(4,5)P2 potentiates eukaryotic Kir channels, its exquisite specificity to phosphate substituents at positions 4 and 5 on the indole ring emphasised by the inability of the PI(4,5)P2 regioisomer PI(3,4)P2 to significantly potentiate activity in all but Kir6 (K$_{ATP}$) channels[12]. While prokaryotic Kir homologues do not share the complete basic sequence required for PI binding, a shorter Arg-Pro motif mediates analogous interactions with cellular phospholipids[13].

It has been assumed that phosphoinositides exert their regulatory effects against a backdrop of the conventional gating model of conformational widening and narrowing of the pore. However, as Kir channels operate and gate without requiring a major conformational change[4], we have sought an explanation for how ion flow through these channels is controlled. A clue is in the location of the regulatory phospholipid binding site, which straddles the membrane-cytosol interface. Structures show that phosphates of the inositol ring of PI(4,5)P2 interact with the Arg-Pro-basic motif[10,14] and neighbouring polar residues. Lipid binding thus creates an extrinsic nexus connecting the transmembrane and intracellular domains. Binding of the PI(4,5)P2 head group is necessary for activation of Kir channels[15] but the accompanying subtle structural remodelling at the membrane-cytosol interface evident in crystal structures offers no cogent explanation for the fine control over K$^+$ flux. We have utilised the prokaryotic Kir channel KirBac3.1 to determine whether the acyl tails of bound lipids, protruding into the conduction pathway through fenestrations just below the selectivity filter, might represent key gating elements.

Here we present evidence that in Kir channels, a limiting barrier to permeation within the conduction pathway is formed by a collar of branched aliphatic side chains. We show that acyl tails of tightly bound lipids occupying conserved fenestrations into the pore engage individual side chains of the collar, drawing them away from the central conduction pathway and allowing ions to pass.

## Results

**The barrier to K$^+$ permeation is internal.** Several crystal structures of Kir channels reveal an internal occlusion comprising branched aliphatic side chains (Val, Ile, or Leu) pointing into the conduction pathway, the location along the molecular axis varying with subfamily (Supplementary Fig. 1). In KirBac3.1, the four Leu124 residues form a steric cluster physically occluding the conduction pathway, separating the cavity into upper and lower vestibules, while Tyr132 (at the inner helix bundle) forms a non-occlusive hourglass-shaped portal between lower vestibule and cytosol (Fig. 1a, b). As ion conduction is not gated at the Tyr132 collar, we considered the possibility that the Leu124 collar might provide the limiting barrier to K$^+$ movement.

Reasoning that the chemistry of the contributing side chains determines their mutual affinity and propensity to form a steric barrier, we mutated the branched aliphatic residue Leu124 to the linear methionine, which is similar in both molecular volume and hydrophobicity, but more flexible. For comparison, we mutated the aromatic Tyr132 at the intracellular entrance to the branched aliphatic residue isoleucine. To evaluate the effect of the mutation on permeation, we reconstituted the L124M and Y132I mutant channels into liposomes and screened activity using a 9-Amino-6-chloro-2-methoxyacridine (ACMA) population assay (Fig. 1c), where a decrease in fluorescence due to protonation of the ACMA fluorophore reflects K$^+$ conduction through liposome-reconstituted channels[16]. Mutation of Tyr132 to isoleucine (Y132I) reduced the magnitude of the fluorescence change, indicating diminished activity. Conversely, L124M assays showed enhanced channel activity, significantly outperforming KirBac3.1 in the fluorometric assay (Fig. 1d). Thus, at either collar, branched aliphatic residues presented an impedance to K$^+$ permeation.

We determined a crystal structure of L124M in case the increased activity was due to structural distortion (such as forcing the inner helices apart) (Supplementary Table 1). The mutation was made on a background in which two native cysteines were mutated to serine (C71S-C262S; SCS) and thus we also determined a structure of the KirBac3.1-SCS background crystallised under the same conditions. The structures superimposed closely (rmsd = 0.33 Å for 100 Cα over the pore and slide helix, and 0.46 Å over all Cα), exhibiting only a minor deviation of the inner helix conformation (Supplementary Fig. 2a–c), whilst maintaining the constriction at Tyr132 (Supplementary Fig. 2d) and the integrity of the selectivity filter[17] (Supplementary Fig. 2e). We also carried out single-channel recordings of the reconstituted L124M channels in lipid bilayers, revealing currents exhibiting the sub-conductance states typifying KirBac3.1[13] (Supplementary Fig. 3).

To delve deeper into the factors underlying the effect of the Leu124 collar on permeation, we investigated the energetic barrier encountered by potassium ions at the Leu124 collar relative to Tyr132 using all-atom enhanced sampling molecular dynamics (MD) simulations across the Leu124 collar from the upper vestibule toward the cytosol (Fig. 1e, Supplementary Fig. 4, Supplementary Table 2). The height of the free energy barrier (potential of mean force; PMF) at Leu124 is ~13 kJ mol$^{-1}$, higher than that of 6 kJ mol$^{-1}$ at Tyr132 but still not a significant barrier to ion passage[18]. Perplexed by the absence of a barrier capable of preventing ion conduction, we carried out simulations on in silico mutants at both constrictions for comparison, focusing on the region of the conduction pathway between the lower vestibule and cytosol. A control Y132F mutant yielded a PMF of 5 kJ mol$^{-1}$ at the phenylalanine collar, similar to the native tyrosine, whereas the isoleucine collar in Y132I created a higher energetic barrier to permeation (PMF = 33 kJ mol$^{-1}$) (Supplementary Fig. 4), consistent with the lower activity of Y132I in ACMA assays (Fig. 1d) and more in keeping with the PMF we had expected at the Leu124 collar. We also carried out simulations on the L124M mutant, using the crystal structure as a starting model, finding a negligible barrier to K$^+$ permeation at the collar.

**Acyl tails of lipids attract and engage aliphatic sidechains.** We had anticipated a large energetic barrier at the leucine collar at residue 124 and were surprised to find a PMF only slightly higher than at the Tyr132 collar. We conducted and analysed the unrestrained simulation data for factors that might explain this. All structures capturing K$^+$ passing the Leu124 collar were extracted from MD simulations and analysed. In every instance,

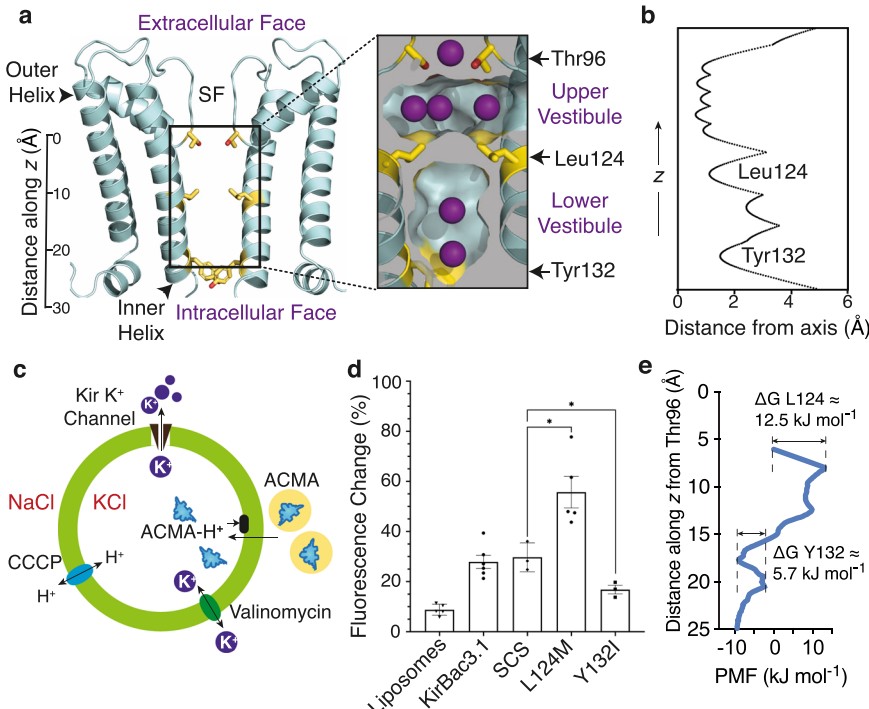

**Fig. 1 Branched aliphatic side chains form a barrier to ions within the conduction pathway. a** Ribbon representation of the KirBac3.1 pore (cyan); for clarity, only two subunits are shown. The side chains of Leu124, Tyr132 and Thr96 are depicted (yellow sticks). A scale bar indicates the distance (Å) from the centre of mass of Thr96. SF indicates the location of the selectivity filter. Insert shows a longitudinal section depicting the accessible surface of the pore interior, highlighting the steric barrier provided by the Leu124 cluster. Potassium ions at binding sites are depicted in purple. **b** Profile of the pore radius through the transmembrane domain of KirBac3.1 calculated using HOLE[56]. Distances are relative to the molecular axis. **c** Schematic of the ACMA assay. A lipid-soluble proton-sensitive dye (ACMA) is equilibrated with proteoliposomes (green circle) in isotonic solution ($K^+_{in}$/$Na^+_{out}$), prior to addition of the protonophore, carbonyl cyanide m-chlorophenyl hydrazone (CCCP). Protons moving into the liposome via CCCP bind ACMA to form ACMA-H$^+$, which does not fluoresce, resulting in a decrease in total fluorescence emission. Proton influx is balanced by $K^+$ efflux through Kir. Protonated ACMA cannot pass out through the membrane. Limiting fluorescence is determined by addition of the specific $K^+$ ionophore valinomycin. The total fluorescence change measured in the assay is summed from individual proteoliposomes. **d** Summary functional assay data for the Leu124M ($n = 5$ independent samples) and Tyr132I ($n = 3$) point mutants represented as mean ± SEM. Values of n for control samples are 5, 6 and 3 for liposome-only, KirBac3.1 and KirBac3.1-SCS samples respectively. Data analysis was by two-sided t-test analysis of variance and refers to pairwise comparisons to KirBac3.1-SCS as indicated (ns = non-significant; *$p \leq 0.05$). Comparative p-values for KirBac3.1-SCS to Y132I and L124M are 0.265 and 0.249, respectively. Dunnett's test was applied to adjust for multiple comparisons to each control. **e** PMF along the molecular axis, oriented to match the inset in (**a**). Distances along z are relative to the centre of mass of the four Thr96 sidechains. Source data are provided as a Source data file.

we observed extended fatty acyl chains of lipids occupying fenestrations near the midpoint of the lipid bilayer, a phenomenon also observed in crystal structures (Fig. 2a, Supplementary Fig. 2f). The carbon chains moved naturally into the fenestrations during the initial MD equilibration steps. Significantly, each $K^+$ permeation event occurred in response to interaction of the fatty acid termini with Leu124 (Fig. 2b). A combination of mutual aliphatic attraction and lipid movement drew the leucine side chains away from the conduction pathway, resulting in limited dissociation of the tetrameric Leu124 cluster and creating an opening of sufficient diameter to allow $K^+$ to pass (Fig. 2c, Supplementary Fig. 5a).

Ions slipped past the Leu124 collar when the mean distance across the opening is approximately 5 Å in diameter (van der Waals' accessible surface) (Fig. 2d, Supplementary Fig. 5a, b), or 40–50 Å$^2$ in cross-sectional area (Supplementary Fig. 5c, d), each $K^+$ retained a coordination shell of four to five water molecules as it passed through (Fig. 2e). Figure 2d shows a density of approximately 35,000 frames/bin for an aperture at the Leu124 collar of sufficient size for conduction; the comparable density when an ion is simultaneously passing is far less, at approximately 250 frames/bin. This resolves any issue of whether the ion pushing past is the cause of collar opening; the Leu124 collar opens as a consequence of lipids

engaging Leu124 residues (Supplementary Movie 1). We note that the interaction of lipidic tails in the fenestrations with Leu124 has negligible effect on the fold of the pore, with rotamer changes of L124 accommodated by small (<1 Å) local perturbations of the inner helix. The action of aliphatic lipid termini in disrupting the tight Leu124 collar thus rationalises the lower-than-expected energetic barrier to $K^+$ permeation.

**Lipid occupancy of fenestrations determines conduction.** To analyse the impact of fenestration lipids on the Leu124 collar, we calculated the probability of the number of lipid termini within 5 Å of Leu124 side chains over all simulation frames (Fig. 3a), observing up to five lipid termini (a few examples had two lipid termini occupying a fenestration) per tetrameric channel. The highest probability in the distribution corresponded to two lipids per tetramer. Probability density profiles of the cross-sectional area at the L124 collar revealed that while even a single fenestration occupied by lipid engaging Leu124 conferred a small probability of an opening of sufficient breadth to allow potassium to pass the collar (Fig. 3a), three or more lipid termini engaging Leu124 greatly increased the probability of achieving a collar area of sufficient size to permit conduction.

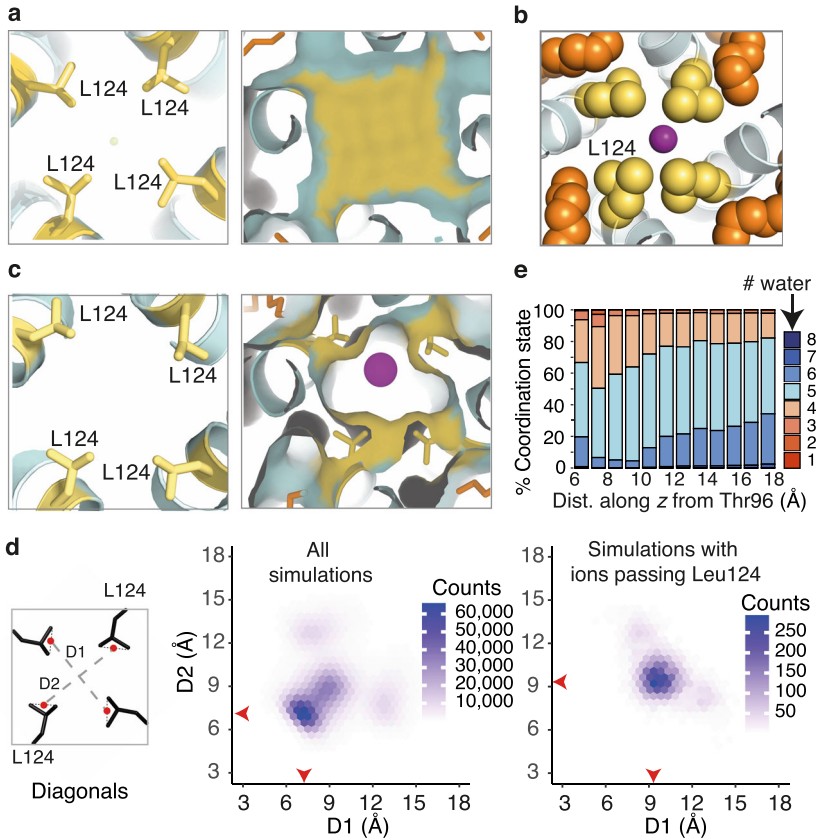

**Fig. 2 Competition for the aliphatic side chains of Leu124 by extended alkyl chains in the fenestrations is sufficient to disrupt the leucine collar and allow K$^+$ to pass. a** (Left) Cross-sectional slice through the KirBac3.1 pore (cyan) showing the central Leu124 side chains. (Right) Accessible molecular surface demonstrating occlusion of the conduction pathway at the Leu124 collar. Side chains and surface of Leu124 are coloured yellow, lipidic alkyl chains orange, and potassium depicted as purple spheres. **b** Close-up representation showing the interaction of Leu124 (yellow spheres) and fenestration lipid (orange spheres). **c** Comparable region of a representative simulation structure extracted as a K$^+$ ion passes the leucine collar. **d** Hexagonal bin plots of D1 against D2 (aperture diagonals) for structures extracted from unbiased non-equilibrium MD simulations. Distances are between points (red dots in the schematic) midway between C$\delta$1 and C$\delta$2. Red arrows on the plots indicate the population of highest density. (Left) Analysis of all structures. (Right) Analysis of those structures in which a K$^+$ is passing the collar. **e** Normalized histograms of data extracted from umbrella sampling simulations enumerate the number of (water) oxygen atoms coordinating each conducted K$^+$ ion as a function of linear distance from Thr96. Each column represents the number of oxygen atoms within 3.0 Å of K$^+$, expressed as a percentage. Source data are provided as a Source data file.

To determine the effect of excluding lipid tails from the fenestrations on the energetic barrier to permeation at the Leu124 collar, we conservatively occluded the fenestrations by altering a neighbouring tyrosine (Tyr57) from the *trans* rotamer observed in the crystal structure ($\chi_1$ (N-C$\alpha$-C$\beta$-C$\gamma$) = 180°) and constraining it in a *gauche* ($\chi_1 = -60°$) conformation; a naturally occurring but low frequency rotamer in KirBac3.1 simulations (Supplementary Fig. 6a). The number of *gauche* rotamers affected width of the aperture at the Leu124 collar; with a tight distribution of occluded pores when all four Tyr57 were *gauche*, and an increasingly variable shape as the number of *trans* rotamers increased (Supplementary Fig. 6b). Importantly, no K$^+$ ions passed the Leu124 collar in simulations where all four fenestrations were blocked by a *gauche* conformation of Tyr57. In this case, the energetic barrier faced by K$^+$ at the Leu124 collar was a prohibitive 41 kJ mol$^{-1}$, substantially greater than the energetic barrier of 13 kJ mol$^{-1}$ observed when lipids are allowed to naturally interact with Leu124, and even greater than that observed in the Tyr132 collar isoleucine mutant Y132I (Fig. 1d, Supplementary Fig. 4). It is thus the action of lipid tails on Leu124 that dissipates the energetic barrier and allows permeation.

We reasoned that if the leucine collar is the primary permeation gate and the role of the lipid tails is to competitively displace individual leucine side chains, thereby compromising the collar, lipids should not be required for permeation in the L124M mutant with its labile methionine sidechains. We thus simulated the L124M mutant under the same conditions as KirBac3.1, excluding lipid from all four fenestrations (4 × Tyr57 *gauche*) for comparison. Indeed, L124M with all four Tyr57 adopting *gauche* rotamers appeared as effective at conducting K$^+$ as wild type KirBac3.1 with all Tyr57 side chains *trans* (Supplementary Fig. 6), indicating that the interaction with aliphatic lipid tails in wild type is only required because of the inflexible steric occlusion created by Leu124 residues. The branched aliphatic leucine side chains comprising the native Leu124 collar thus form a gate-able barrier to conduction, operated by chemically similar aliphatic lipid chains in a competitive tug-of-war.

To cross-validate this experimentally, we mimicked isolated lipid tails using alkyl-methylthiosulfonate (alkyl-MTS) reagents, derivatising a native cysteine residue (Cys119) located near the midpoint of the bilayer at the outer opening to the fenestration on an otherwise cysteine-less background (SCS); the same background as used for L124M. All Cys119 MTS-derivatives have an intact Leu124 collar (*i.e.* not L124M). Derivatisation was verified by gel shift assay of samples after reacting unlinked cysteines with 5 mM methoxy PEG maleimide (Supplementary Fig. 7). Two reagents, the substituents differing by four carbons in length, were employed; reasoning decyl-MTS should be sufficiently long to

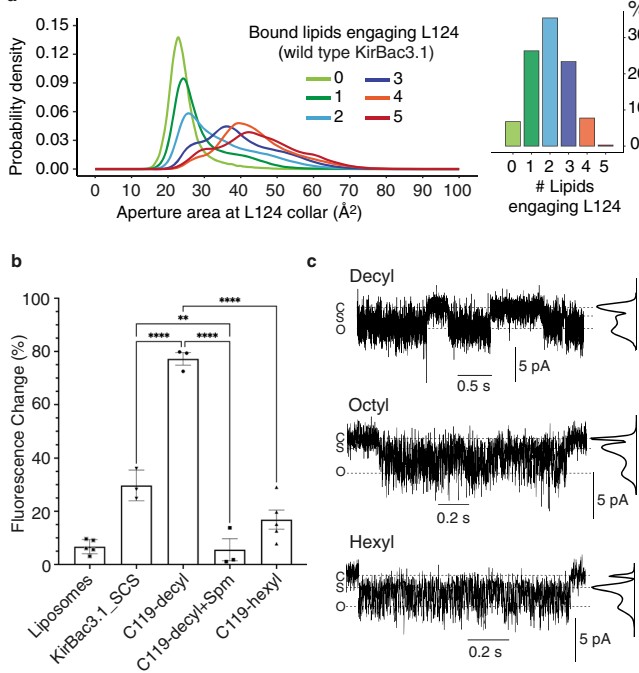

**Fig. 3 Alkyl chain length and relative occupancy determine collar opening and channel activity. a** The number of lipids engaging with Leu124 side chains varies as a normal distribution. Its relationship to the cross-sectional area of the open collar between leucine side chains is depicted. **b** Summary ACMA assay data for the C119-decyl ($n = 3$ independent samples) and C119-hexyl derivatives ($n = 5$ independent samples) represented as mean ± SEM. Values of n in control samples are 5, 3, and 3 for liposomes only, KirBac3.1-SCS, and C119-hexyl+spermine. Data analysis was by two-sided $t$-test analysis of variance and refers to pairwise comparisons to KirBac3.1 or KirBac3.1-SCS as indicated (ns = non-significant; $*p \leq 0.05$; $**p \leq 0.01$; $***p \leq 0.001$; $****p \leq 0.0001$). Comparative $p$-values for comparison of C119-decyl to C119-hexyl, C119-decyl+spermine, and KirBac3.1-SCS are all <0.0001. The $p$-value for comparison of KirBac3.1-SCS to C119-decyl+spermine is 0.0037. The 95% confidence intervals, in respective order, are (46.1–74.6), (55.8–87.5), (31.6–63.4), and (8.9–39.4). Dunnett's test was applied to adjust for multiple comparisons to each control. Source data are provided as a Source data file. **c** Single channel recordings of C119-decyl, C119-octyl and C119-hexyl at −80 mV, filtered at 500 Hz. Open (O), closed (C) and partially open substate (S) levels determined by Hidden Markov Models are indicated by horizonal dashed lines. Maximum-likelihood amplitude histograms generated using Hidden Markov Models are aligned with the recordings.

fully engage Leu124, while hexyl-MTS should not. We hypothesised that at high occupancy, hexyl chains would block the fenestrations, diminishing activity if lipids are mediators of conduction but leaving activity unaffected if they are not. ACMA fluorimetric population assays on the reconstituted derivatised channels showed bound decyl chains (C119-decyl) increased activity relative to wild type, consistent with Leu124-lipid interaction, while the hexyl reagent (C119-hexyl) reduced function (Fig. 3b, Supplementary Fig. 8, Supplementary Tables 3, 4), consistent with exclusion of membrane lipid. The data indicate that ion permeation is contingent upon lipid tails being of sufficient length to reach the occlusion.

To directly compare the effect of alkyl derivatisation on the activity of individual channels, we performed electrophysiological recordings in artificial bilayers on channels derivatised with one of three MTS-reagents to titrate the effect of chain length; C119-decyl, C119-octyl and C119-hexyl. Current–voltage (I–V) curves

from single channel recordings of the spermine-sensitive channels show the decyl, octyl and hexyl-derivatised channels remain fundamentally similar (Supplementary Fig. 9). Analysis of the recording data using Hidden Markov Modelling shows that C119-decyl has a much higher probability of operating at maximal conductance during activation bursts (mean open probability, Po = 0.69) than C119-octyl (Po = 0.41) or C119-hexyl (Po = 0.46) and is more likely to be fully open than operating in a lower amplitude substate (Ps). Substitution with the shortest alkyl chain, C119-hexyl, yields a channel most likely to be operating at a lower conductance substate level, while C119-decyl is the least likely to operate at substate level (Fig. 3c, Supplementary Fig. 9). We note that relative to the C119-derivatised channels, the L124M mutant has the highest probability of being fully open, consistent with its ability to function in the absence of fenestration-bound lipid (Supplementary Fig. 3).

To verify that the C119-decyl substituents were indeed emulating the effect of lipid, we ran unrestrained MD simulations as before, estimating the probability of fenestration occupancy by decyl substituents and the corresponding effect on the Leu124 aperture. As lipid tails were also occasionally observed in fenestrations, the total number of aliphatic chains entering the fenestrations were enumerated. Most usually, the C119-decyl tail engages a Leu124 side chain (i.e. lies within 5 Å of any atom of Leu124) in three of the four fenestrations (Fig. 3c, Supplementary Fig. 10a–c). The increase in occupancy relative to wild type channels (Fig. 3a) may be due to covalent binding of the decyl substituents at the fenestration entrance. In C119-hexyl, by contrast, the hexyl chains partially enter the fenestrations but rarely far enough to engage Leu124; the highest probability is at zero (no engagement) with about 3% of structures with a single Leu124 engaged by a hexyl substituent. The more likely outcome is displacement of hexyl substituents by lipid termini from the bilayer (~19 %). (Supplementary Fig. 10d–f). Both scenarios can result in a breach of the steric cluster, but very few ion permeation events are observed.

**Anionic lipids are tightly retained by the channel**. Our functional experiments indicated that, in the same manner as human Kir channels are potentiated by phosphoinositides, the prokaryotic Kir channel has a requirement for anionic lipids. To investigate the requirement for anionic lipid head groups, we carried out 100 μs coarse-grained MD simulations on KirBac3.1 embedded in membranes comprising a 1:1:1 ratio of phosphatidylcholine:phosphatidylserine: phosphatidylglycerol (PC:PS:PG). We evaluated differences in protein-lipid association by estimating the number density distribution of lipid head groups surrounding the pore in the inner bilayer leaflet (Fig. 4a). The PC head groups were broadly distributed, forming a diffuse halo at a radial distance of 25–45 Å from the principal molecular axis, which defines the centre of the simulation system. The heads of the anionic lipids exhibited a much tighter distribution, enriched at a radius of ~25–30 Å, which is comparable to the distance between the PI(4,5)P2 head group at the canonical lipid binding site and the molecular axis in structures of Kir2.2 (e.g. PDB entry 3SPI)[10]. Coupled with evidence of 4-fold symmetry in the distribution of the anionic lipids, the data suggest PS or PG are likely to occupy the canonical lipid binding site, whereas PC loosely associates with the channel. Lipids in the canonical binding site are better placed to occupy the surface cleft connecting each binding site with a fenestration (Fig. 4b, Supplementary Fig. 11).

For the purpose of clearly defining any relationship between the fenestration-bound lipid tails and head group binding, we enumerated the proximity of the phosphate of the lipids to the

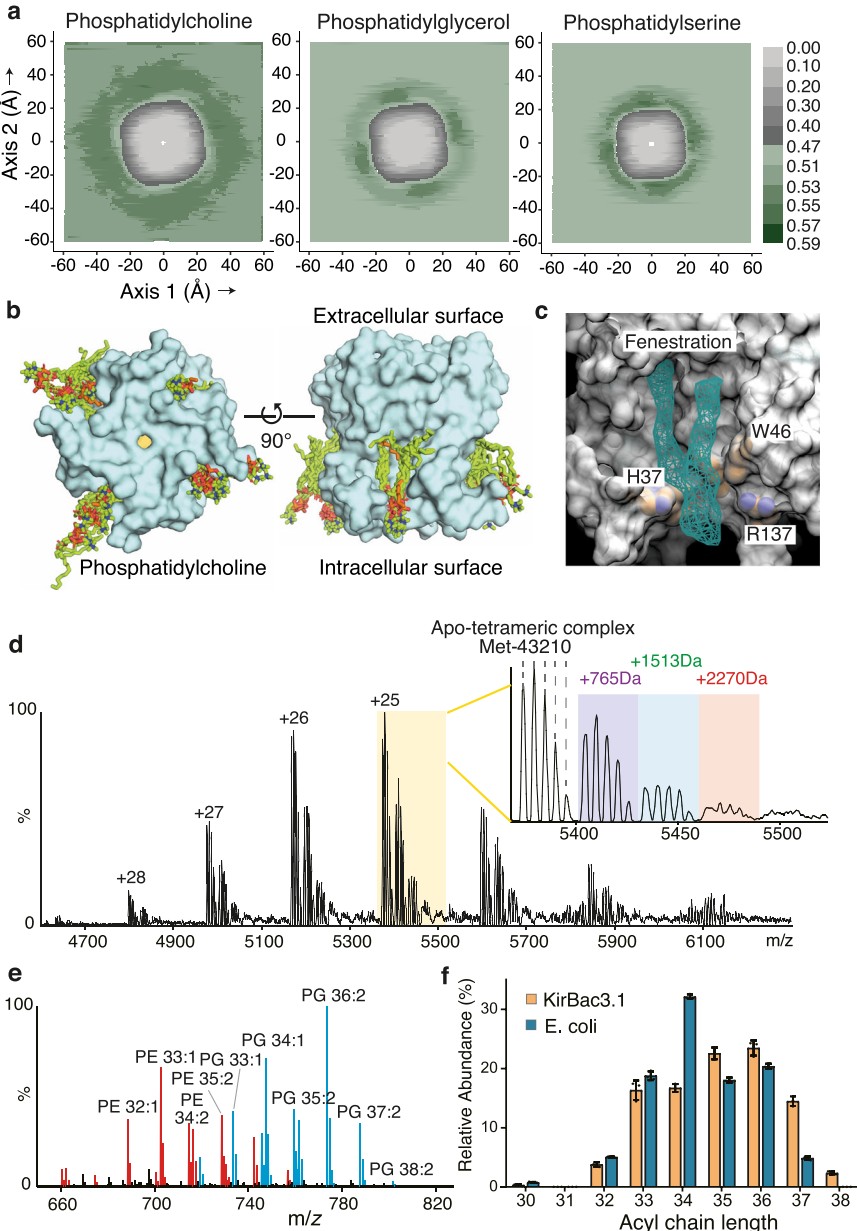

**Fig. 4 Anionic phospholipids bind tightly and specifically to the pore. a** Projections along z of coarse grained simulation data showing the number density distribution of lipid head groups surrounding the pore. Source data are provided as a Source data file. **b** The position of protein associated POPC lipids from 10 randomly selected conducting structures from steered MD superimposed on the crystal structure of the KirBac3.1 pore illustrate a binding cleft in the molecular surface connecting cytosol and fenestrations. POPC lipids are depicted as green sticks, whereas the lipidic fragments depicted in orange arise from the crystal structure. **c** An isosurface of the fenestration-bound lipids from these calculations demonstrates the relationship of lipid to a canonical lipid-binding site; the isosurface was calculated from 292 lipid molecules at a mass density contour level of 0.15. The isosurface is shown as teal mesh, and sphere representations of the His, Arg and Trp are coloured on an otherwise grey molecular surface. **d** Native mass spectrum of KirBac3.1 protein in 200 mM ammonium acetate (pH = 8.0) and 0.5% C8E4 (2× the critical micelle concentration) reveals tetrameric complexes with up to four N-terminal methionine truncations (Met- 0, 1, 2, 3 and 4), and with one to three phospholipids binding (+765 Da, +1,513 Da and 2270 Da, respectively). **e** Lipidomics analysis of KirBac3.1 suggests that PG is enriched amongst the co-purified lipids. **f** Comparison of the total acyl chain length of PG in KirBac3.1-co-purified lipids and total E. coli lipids suggests that KirBac3.1 preferentially associates with long chain lipids. Error bars represent mean ± SD, with n = 3; individual measurements are overlaid as dot plots. Source data are provided as a Source data file.

lipid binding site using the all-atom MD data. This revealed that even with neutral PC, 95.5 % of the lipids occupying fenestrations had head groups falling within 5 Å of the Arg137, Trp46 and His37 pocket, indicating that lipids occupying fenestrations almost invariably have tightly bound head groups. An isosurface of the fenestration-bound lipids from frames in which the $K^+$ ion is found transiting the Leu124 collar demonstrates the relationship of lipid to a canonical lipid-binding site (Fig. 4c). The

condensation of PS and PG headgroups revealed in coarse-grained simulations suggest additional electrostatic contacts facilitated by anionic lipids in pore and intracellular assembly are critical in attracting and confining PS and PG. The nexus of electrostatic contacts to both pore and intracellular elements of the protein in the vicinity of the tetramer interfaces is a prerequisite for coupling interdomain motions of the Kir channel[13] to the movement of the lipid tails engaging Leu124.

We utilised mass spectrometry methodology to identify the lipids carried through in purification of wild type KirBac3.1 expressed in *E. coli*. The native mass spectrum of purified KirBac3.1 displayed an adduct peak series with an average additional mass of 765 Da, relative to the native tetramer (Fig. 4d). Further lipidomics analysis confirmed the identity of the lipid adducts as phosphatidylethanolamine (PE) and PG, with PG adducts being enriched (Fig. 4e). Examining the fatty acyl chain length of lipids co-purifying with KirBac3.1 revealed enrichment of species with long acyl chains (>34 total carbons over the two acyl chains) over their proportions in *E. coli* lipid mixes, consistent with preferential binding of long chain fatty acids (Fig. 4f).

## Discussion

We previously demonstrated that Kir channels conduct K[+] through a very narrow pore[4], re-opening the question of how ion permeation is controlled in these channels. Here, in a major conceptual shift, we present evidence that lipid molecules tightly associated with Kir channels operate an occlusive barrier between the selectivity filter and cytosolic portal. All crystal structures of native KirBac3.1 show the conduction pathway is occluded by a Leu124 cluster, indicative of a favoured low-energy state.

While the function of many K[+] channels is known to be enhanced by the presence of anionic lipids[19], notably PI(4,5)P2[20], the interaction of Kir channels with anionic lipids is obligate, and binding of the anionic head moiety is essential for activation. It is insufficient for function, however, and here we show that a steric barrier, comprising branched aliphatic side chains, is operated in a stochastic manner by fatty acyl termini. The acyl chains insinuate into the conduction pathway to competitively sequester leucine side chains, compromising the leucine cluster to create an aperture sufficiently wide for K[+] to pass. We observed that potassium ions were coordinated by four to five water molecules as they passed the Leu124 collar. This is higher than the coordination number of ions passing the Tyr132 collar at the inner helix bundle constriction, in which the aperture is correspondingly smaller[4], illustrating a key energetic difference between constrictions comprising branched aliphatic and aromatic side chains; leucine forms a hydrophobic occlusion while tyrosine stabilises partially hydrated K[+] by cation-pi coordination, meaning fewer coordinating water molecules are required to shield the K[+] charge.

We were able to experimentally manipulate the steric occlusion at Leu124 by two means, mutagenesis and lipid mimicry. In simulations, the L124M mutant was able to conduct even when lipid tails were prevented from entering fenestrations, demonstrating the unbranched aliphatic residue methionine does not form a significant barrier to permeation; L124M is essentially ungated. We were able to modulate activity by changing the length of the covalently associated lipomimetic substituents relative to the distance between Cys119 and Leu124, rationalising the preferential binding of lipids with long acyl chains revealed by mass spectrometry analysis.

We have established that the aliphatic termini of lipids are as essential to function as their head groups, and suggest the remote functional moieties, half a bilayer apart, exert allosteric control over conduction. While it is not essential that the remote functions are carried out by the same lipid molecule, under physiological conditions it is most likely to be the case, facilitated by the natural hydrophobic surface crevices connecting the head-binding site to the centre of the bilayer. Indeed, in MD simulations of the pore, fenestration-occupying lipids almost invariably occupy these crevices and their tightly bound head groups make multiple electrostatic contacts to both pore and interdomain

linker in the vicinity of the tetramer interfaces. In the full-length protein, the nexus of contacts across the domain boundary would directly couple interdomain motions[13] to the movement of the lipid tails engaging Leu124, which operate the gate. The resultant connection is potentially capable of providing the control over gating of eukaryotic Kir channels that is required for maintaining neural and organ function in higher organisms, where contacts made by the head group of PI(4,5)P2 link the pore and intra-cellular domains, subtly rearranging the interfacial region[10,13] and potentiating the channel. In the light that intracellular domains of Kir channels move independently, rotating and re-orienting relative to the pore[13,21,22], it is tempting to speculate that, through the extrinsic lipid connections, interdomain motions manifest in the depth of penetration of the lipid termini into fenestrations, and in turn to their capacity to tug a blocking leucine side chain.

We have demonstrated using MD simulations that the key criterion for permeation is widening of the Leu124 collar, and that this is correlated to the number of lipids engaging Leu124 side chains. When lipid entry is hampered by partially or fully blocking the fenestrations (C119-hexyl or Tyr57 *gauche*) we observe fewer, or no, permeation events, respectively. Moreover, the phenomenon of conduction substates in K[+] channels is not well understood, and our collective data hint that the conduction substates we observe in single-channel recordings may have their origins in a fluctuating number of Leu124 side chains engaged by the termini of alkyl chains.

Our findings sit within an emerging context of a general sensitivity of ion channels, including all major classes of K[+] channels, to phosphoinositides[20,23–25] and anionic lipids in general[19]. Analogous fenestrations have been reported in diverse ion channel families, structures including but not limited to TREK-2[26], K2P1[27] and TRIC[28], and experimental data implicating fenestration lipids in TRPC3 activity[29], suggesting the concept of specifically bound phospholipids as allosteric gating elements may be generally applicable.

Here we have introduced the concept that control over permeation in Kir potassium channels is achieved by an interactive relationship between the channel and bound phospholipids, demonstrating that the aliphatic termini of lipids, acting on a steric occlusion within the conduction pathway, lower the free energy barrier to ion permeation. Based on this concept, we have been able to enhance and attenuate activity according to prediction, verifying the nature, site, and operating mechanism of a permeation gate. In a physiological context, we submit that the activity of Kir channels is dependent on a specific association with lipids having fatty acyl substituents of sufficient length to extend from a tightly bound anionic head group at the intracellular surface to the conduction pathway, entering via fenestrations in the pore walls.

## Methods

**Materials**. Detergents are from Anatrace, lipids from Avanti Polar Lipids and all other reagents, unless otherwise stated, purchased from Sigma Aldrich. Kirbac3.1 mutant cDNA was purchased from Genscript. Chromatography columns and resins are from GE Healthcare.

**Protein preparation**. Kirbac3.1 was expressed in C41(DE3) (Lucigen Cat# 60442) *E. coli* at 18 °C for 18 h. Cells were harvested and resuspended in 20 mM TRIS pH 8.0 and 150 mM KCl containing EDTA free Complete Protease Inhibitor (Roche) followed by lysis by two passes through a STANSTED SPCH-10 high pressure homogeniser from Homogenising Systems (UK) at 100 MPa. Membranes were harvested by centrifugation at 265,000 × *g* for 60 min at 4 °C and the protein solubilised in 100 ml of 20 mM TRIS pH 8.0, 150 mM KCl, 1 mM PMSF and 1% (w/v) 3,12-Anzergent for 45 min. Insoluble material was removed by centrifugation. Imidazole to 20 mM was added to the supernatant, which was loaded on a 5 ml IMAC HiTrap column (GE) charged with Co[2+]. Protein was eluted with a linear gradient of imidazole, up to 500 mM, in buffer containing 20 mM TRIS pH 8.0, 500 mM KCl, 0.1 mM PMSF and 0.05% (w/v) dodecyl-β-D-maltopyranoside

(DDM). Fractions containing KirBac3.1 were concentrated in a 100 kDa molecular weight cut-off centrifugal concentrator (Millipore) and applied to a Superdex 200 10/30 column (GE) equilibrated with 20 mM TRIS pH 8.0, 150 mM KCl, 0.02% (w/v) DDM; with 0.5 mM TCEP added to samples that did not require further chemical derivatisation. Fractions corresponding to the KirBac3.1 tetramer were concentrated with a 100 kDa centrifugal concentrator (Millipore) to between 10 and 40 mg ml$^{-1}$, snap frozen in liquid N$_2$ and stored at $-80\,°C$ as single use aliquots.

**MTS derivatisation and evaluation.** KirBac3.1 mutants were diluted to 1 mg ml$^{-1}$ in crosslinking buffer (20 mM TRIS pH 8.0, 150 mM KCl and 0.02% (w/v) DDM) and crosslinked by addition of decyl or hexyl (MTS), to 4 mM or 5 mM, respectively, followed by incubation at room temperature for 2 h with gentle agitation. Excess MTS reagents were removed with Micro Bio-Spin 6 columns (Bio-Rad) equilibrated in crosslinking buffer. The tetramer was re-purified by size exclusion chromatography with a Superdex 200 10/30 column (GE) equilibrated with the crosslinking buffer. Fractions corresponding to the KirBac3.1 tetramer were recovered and concentrated with a 100 kDa centrifugal concentrator (Millipore) to between 10 and 20 mg ml$^{-1}$, snap-frozen in liquid N$_2$ and stored at $-80\,°C$ as single use aliquots until required.

To evaluate the completeness of the reaction we modified the method of Accardi[30]. An aliquot (11 µl) of the reacted protein was denatured by incubation with 44 µl 100% (v/v) acetone stored at $-20\,°C$. The sample was incubated overnight at $-20\,°C$, before pelleting via centrifugation at maximum speed at 4 °C in a benchtop centrifuge. The supernatant was carefully removed, and the pellet air-dried. Methoxy-polyethylene glycol maleimide 5000 (mPEG5K) was made up to 5 mM in 8 M Urea and 2% (w/v) SDS before the protein pellet was resuspended in 10 µl mPEG5K and incubated for 1 hr at room temperature. Samples were then analysed via SDS-PAGE. KirBac3.1 runs as a monomer on SDS-PAGE. Free cysteines result in a gel-shift of the band by the approximate mass of the mPEG5K, whereas derivatised cysteines are protected from PEGylation and migrate at the same rate as the wildtype.

**Crystallization and data collection.** KirBac3.1 SCS mutants at a concentration of 4 mg ml$^{-1}$ were crystallized in sitting drops by vapour diffusion at the Bio21 C3 crystallisation facility in Parkville against 2.5% PEG 4,000; 2.5% PEG 8000; 10–21% PEG 400; 90 mM HEPES pH 7.5, 1 mM TCEP and 50 mM EDTA, or 10 mM CaCl$_2$. The crystals were cryoprotected prior to data collection by the addition of TMAO pH 8.0 to ~2 M. All crystallographic data were collected at beamline MX2 at the Australian Synchrotron using Blu-Ice[31].

**Structure determination and refinement.** Crystallographic data were processed and scaled using XDS[32] and reduced using the CCP4 suite[33]. Electron density in all structures was phased by molecular replacement, using *PHASER*[34] as implemented in the CCP4i suite. A molecular replacement model for the wild type structure was derived from previous protein data bank entries from our group. Electron density maps for the structures were B-factor sharpened. Model building was carried out using COOT[35] and refinement using PHENIX[36]. Atomic coordinates were iteratively refined by maximum likelihood, and simulated annealing procedures applied, alternating positional refinement cycles with individual B-factor refinement. Occupancy was refined for ligands, ions and residues with alternate conformations. Refinement was monitored according to the decrease in $R_{free}$. A number of residues in each structure were omitted or had side chains truncated due to positional disorder. Statistics are presented in Supplementary Table 1.

**Molecular dynamics.** The 2.0 Å crystal structure of native KirBac3.1 (PDB 6O9U)[4] was used as an initial model for wild type, Y132 mutants, C119-decyl, C119-hexyl and Y57-related simulations. The 2.7 Å crystal structure of the L124M mutant was used for L124M simulations. The whole simulation systems were prepared using the CHARMM-GUI Membrane Builder server[37], with N- and C- terminal residues patched as standard termini. Transmembrane pores (residues 33–138) were embedded within a standard POPC bilayer, each containing 162 lipids, using the replacement method. Additionally, each starting model contained five potassium ions: three sites within the selectivity filter (S1, S3, S4) and two in the internal cavity (near the selectivity filter). One additional potassium ion was placed in the lower cavity. Thirty water molecules were also placed in the cavity. The simulation cells comprised approximately 60,000 atoms, of which approximately 30,000 were water molecules, in a box of dimensions $8.2 \times 8.2 \times 8.4$ nm$^3$. TIP3P water parameters were used to solvate all systems and all ionisable residues were assumed to be in their dominant protonation states at pH 7. Sufficient K$^+$ and Cl$^-$ ions were introduced by replacement of water molecules to bring the systems to an electrically neutral state at an ionic strength of 0.15 M. The principal axis of the pore was aligned along the z-axis.

All simulations were performed using the GPU-accelerated GROMACS software package (version 2020.1)[38] and CHARMM36m force field[39]. CMAP corrections were applied for all components; all atoms including hydrogen atoms were included. CGenff[40] program was used to parameterize the C119-decyl and C119-hexyl residues. Dihedral restrains were applied onto the dihedral N-CA-CB-CG on Y57 at $-65 \pm 10°$ with harmonic force constant of 1000 kJ mol$^{-1}$ nm$^{-2}$ in

Y57-related simulations to maintain a *gauche* rotamer. To obtain the equilibrated channel-ion-solvent-membrane complex, an elongated CHARMM-GUI Membrane Builder preparation protocol was followed. Steepest descent energy minimization was followed by six sequential steps of equilibration with a gradual decrease in the restraining force applied to different components. After 101.5 ns NVT and NPT equilibrations, the resulting structures were utilized in steered molecular dynamics (SMD) or directly in unrestrained MD. The LINCS[41] algorithm was applied for resetting constraints on covalent bonds to hydrogen atoms, which allowed 2 fs time steps for MD integration during the entire simulation. The particle-mesh Ewald[42] algorithm was used for calculating electrostatic interactions within a cut-off of 12 Å, with the Verlet grid cut-off-scheme[43] applied for neighbour searching, using an update frequency of 20 and a cut-off distance of 12 Å for short-range neighbours. A 12 Å cut-off was applied to account for van der Waals interactions, using a smooth switching function starting at 1.0 nm. Periodic boundary conditions were utilized in all directions. During the equilibration stages, the temperature was maintained at 303.15 K using a Berendsen-thermostat[44] with a time constant of 1.0 ps. Protein, membrane and ion-water groups were treated independently to increase accuracy. The pressure was maintained at 1.0 bar by semi-isotropic application of a Berendsen-barostat[44], with a time constant of 5.0 ps. During steered molecular dynamics, the temperature was maintained at 303.15 K using a Nose–Hoover-thermostat[45] with a time constant of 1.0 ps, with protein, membrane and ion-water groups treated independently, with the pressure maintained at 1.0 bar using the Parrinello–Rahman-barostat[46] semi-isotropically, with a time constant of 5.0 ps and compressibility of $4.5 \times 10^{-5}$ bar$^{-1}$. The umbrella sampling method was used to calculate the potential of mean force (PMF) before ions passed through the leu124 collar until fully into solvent. The reaction coordinate of a permeating ion was defined relative to the center-of-mass (COM) of the four Thr96 residues at the base of the selectivity filter along the z direction. The initial structures for umbrella sampling were extracted from the pulling SMD trajectory. Pulling potentials were set for two discrete pulling groups: the potassium ion in the upper cavity and four Thr96. The pulling geometry was direction-periodic, with a harmonic force constant during umbrella pulling of 3000 kJ mol$^{-1}$ nm$^{-2}$. The rate of change was 0.2 Å ns$^{-1}$ in each case. Although subjecting a target ion to pulling forces, a flat-bottomed position restraint was applied to the ion to restrain it within a cylinder with a radius of 1 Å running parallel to the z axis. Each umbrella sampling window was simulated for 310 ns or 200 ns dependant on the total sampled length of reaction coordinates. The windows were centred at ~0.5 Å intervals along the z axis, covering the range from 16.5 to 36.5 Å relative to the centre-of-mass of the four Thr96 residues; during umbrella sampling the flat-bottomed position restraint on pulled ion was removed. A static external electric field with a strength of 0.05 V nm$^{-1}$ was applied along the membrane normal direction toward the cytoplasmic domain side in steered molecular dynamics and umbrella sampling. Final PMF profiles were calculated using gmx-wham[47] and the statistical uncertainties estimated via Bayesian bootstrap analysis with 200 bootstraps and a tolerance of $10^{-6}$.

The number of water oxygen atoms within a prescribed cut-off distance of 3.0 Å of the target ion along traveling reaction coordinate was calculated using VMD[48]. For unrestrained MD, the number of lipids and/or C119-decyl/hexyl alkyl chains within cut-off distance of 5.0 Å of the sidechain atoms of Leu124 was also calculated using VMD. All probability density profiles and two-dimensional histograms were constructed using the ggplot2[49] package in the R software package.

All unrestrained simulations were carried out with a static external electric field with a strength of 0.05 V nm$^{-1}$ applied along the membrane normal direction; a potential across the simulation box of approximately 420 mV operates across the membrane[50]. Each simulation was run over 200 ns. The interchangeable diagonals, edges, area were statistically analysed as D1 and D2 couplets for structures extracted with 10 ps intervals. All hexbin figures were symmetrized by exchanging the indices of diagonals; that is, the couplets of (D1, D2) were duplicated in the form (D2, D1). All probability density functions were calculated using the ggplot2 package in R software and integration calculations were performed the Grace plotting tool.

Coarse-grained simulations were performed using the GPU-accelerated GROMACS software package (version 2020.1)[38] and martini22 force field[51]. Martini2.2 parameters were using to treat protein section, and Martini2.0 parameters for lipids and ions and non-polarizable water. CHARMM-GUI Martini Bilayer Maker was used to construct simulations system. Transmembrane pores (residues 33–138) were embedded into lipid bilayer has 34% POPC, 33% POPG and 33% POPS in both leaflets. 800 ns NVT and NPT equilibration steps with a gradual decrease in the restraining force on channel atoms were taken prior the each 20 µs production MD. Five replicates in total were run. The gmx-densmap program was used to calculate the planar distributions of lipid head groups.

Summarised experimental parameters can be found in Supplementary Table 2.

**KirBac Liposome reconstitution.** Lipids (50% 1-palmitoyl-2-oleoyl-sn glycero-3-phosphocholine:POPC, 5% 1-palmitoyl-2-oleoyl-sn glycero-3-phospho-L-serine:POPS, 45% total *E. coli* lipids:TECL in the ratio POPC:TECL:POPS 50:45:5) were dissolved in chloroform:methanol (65:35), dried to a thin film under N$_2$ and residual solvent removed by lyophilization overnight. Water was added to a final

lipid concentration of 25 mg ml$^{-1}$ followed by incubation at 37 °C for 1 h and 5 freeze/thaw cycles. An equal volume of 2× inside buffer (IB) was added to give a final buffer composition of 20 mM Tris-HCl pH 8.5 and 150 mM KCl followed by extrusion to 400 nm (two rounds of 11 passages through the filter) in a mini-extruder (Avanti Polar Lipids) preheated to 65 °C. In some experiments NaCl was substituted for KCl.

For reconstitution, dodecyl-β-maltopyranoside (DDM) was added to a 1:8 molar detergent:lipid ratio to liposomes and incubated for 15 min at room temperature followed by the addition of protein to a protein:lipid molar ratio of 1:555 and incubation for 1 h at room temperature.

Excess detergent was removed by passing 80 μl of proteoliposomes through G50 resin (settled bed volume 3 ml) equilibrated in inner buffer (IB), followed by elution with IB. Further detergent removal was accomplished by dilution of liposomes to a final volume of 1.5 ml by centrifugation at 91,000 × g for 1 h at 25 °C. The liposome pellet was resuspended in 100 μl IB, centrifuged at 3000 × g for 3 min and any precipitate discarded. Liposome size uniformity was assessed using dynamic light scattering (Zetasizer – Malvern Instruments) and protein reconstitution by SDS-PAGE using known concentrations of KirBac3.1 in DDM as standards. Amount of protein added was optimized using wild type KirBac3.1 to achieve an average of 0.3 functional channels per liposome, with equivalent protein:lipid ratios used for the crosslinked experiments.

**Liposomal fluorimetric assay.** Fluorimetric liposome assays were performed within 24 h of KirBac reconstitution. Each 5 μl aliquot of prepared liposomes was diluted to a final volume of 110 μl in outside buffer (OB: 20 mM Tris-HCl pH 8.5 and 150 mM NaCl), resulting in a final buffer composition of 20 mM Tris-HCl pH 8.5, 143.2 mM NaCl and 6.8 mM KCl. The pH sensitive dye 9-amino-chloro-2-methoxyacridine (ACMA) was added to a final concentration of 2 μM from a freshly prepared 200 μM stock in 100% (v/v) ethanol. Fluorescence emission at 483 nm (excitation at 419 nm) was monitored over time with path lengths of 2 and 10 mm for excitation and emission, respectively. Baseline fluorescence was monitored for 3 min before the proton specific ionophore carbonyl cyanide m-chlorophenyl hydrazine (CCCP) was added ($t = 0$) to a final concentration of 9 nM from a freshly prepared 500 nM stock prepared in OB. A decrease in fluorescence is observed when the membrane is permeable to K$^+$. After 35 min ($t = 35$), the K$^+$ specific ionophore was added to 20 nM from a freshly prepared 2 μM stock made up in 100% (v/v) ethanol. The assay was allowed to reach a final electrochemical equilibrium over a further 20 min. In some experiments, the K$^+$ channel blocker spermine was added to a final concentration of 500/1000 μM in IB/OB before the establishment of the fluorescence baseline. Control liposomes were prepared identically to proteoliposomes with the exception of the addition of an equal volume of crosslinking buffer in place of protein.

All fluorescence experiments were normalized to the fluorescence measured at $t = 0$, which represents maximal fluorescence (100%); and the fluorescence measured after the addition of valinomycin, which represents the minimal fluorescence where all liposomes have equilibrated (0%). A first-order exponential decay was fitted from $t = 0$ to $t = 35$ to determine the change in fluorescence due to K$^+$ flux through reconstituted KirBac channels.

**Single channel measurements.** KirBac3.1 proteoliposomes were incorporated into planar lipid bilayers separating two aqueous chambers (*cis* and *trans*). Bilayers were produced using the Mueller method[52]. Bilayers were formed across an aperture in a delrin cup with diameter 150–250 μm using a lipid mixture of phosphatidylethanolamine, phosphatidylcholine and phosphatidylserine (7:2:1 wt/wt, Avanti Polar Lipids) in *n*-decane (50 mg ml$^{-1}$, ICN Biomedicals). The *cis* chamber contained 250 mM KCl and 1.0 mM CaCl$_2$ and the *trans* chamber contained 50 mM KCl and 0.1 mM CaCl$_2$. Proteoliposomal fusion with bilayers was initiated by adding 5 μl aliquots of the proteoliposome preparation (described above) to the *cis* bath whilst stirring. During experiments, the composition of the *cis* solution was altered by a perfusion system[53] that allowed exposure of single channels to additions of spermine or KCl/NaCl substitutions within ~1 s. All solutions were pH buffered using 10 mM N-tris[hydroxymethyl] methyl-2-aminoethanesulfonic acid (TES; ICN Biomedicals) and titrated to pH 7.4 using KOH (ICN Biomedicals). KCl was obtained from Aldrich and CaCl$_2$ from BDH Chemicals. Experiments were carried out at room temperature (23 ± 2 °C).

**Data acquisition and analysis.** Electric potentials expressed as *trans* relative to *cis* at virtual ground. Control of the bilayer potential and recording of unitary currents was done using an Axopatch 200B amplifier (Axon Instruments Pty, Ltd). Channel currents were digitized at 50 kHz and low pass filtered at 5 kHz using a data interface (Data Translation DT301) under the control of in-house software written in Visual Basic version 6. Before analysis the current signal was re-digitized at 1 kHz and low pass-filtered at 1 kHz. Channel substate analysis was carried out using the Hidden Markov Model (HMM)[54]. The algorithm calculates from the raw signal the most likely amplitude histogram, the idealised, three-level, current time course with background noise subtracted and the transition-rate matrix using maximum likelihood criteria.

**Native mass spectrometry.** KirBac3.1 protein was buffer-exchanged to 200 mM ammonium acetate (pH = 8.0) with 0.5% C8E4 (2×CMC) using a Biospin-6 column (BioRad). The protein sample was loaded into a gold coated needle prepared in-house and analysed on a modified Q-Exactive mass spectrometer (Thermo Fisher Scientific). The typical MS settings were spray voltage of 1.3 kV, source fragmentation of 50-150 V, HCD collision energy of 50–200 V and resolution of 17500 at *m/z* 200.

**Lipidomics.** KirBac3.1 co-purified lipids and *E. coli* total lipids were analysed using mass spectrometry based lipidomics[55]. Briefly, protein was digested with trypsin overnight at 37 °C, lyophilized and re-dissolved in 70% solution A (acetonitrile:H$_2$O 60:40, 10 mM ammonium formate and 0.1% formic acid) and 30% solution B (isopropanol:acetonitrile 90:10, 10 mM ammonium formate and 0.1% formic acid). The lipids were loaded onto a C18 column (Acclaim PepMap 100, C18, 75 μm × 15 cm, Thermo Scientific) by a Dionex UltiMate 3000 RSLC Nano system coupled to an LTQ Orbitrap XL hybrid mass spectrometer (Thermo Fisher Scientific). The lipids were separated with a gradient from 30% buffer B to 99% buffer B. The LTQ-Orbitrap XL was set up in negative ion mode and in data-dependent acquisition mode to perform five MS/MS scans per MS scan. Survey full-scan MS spectra were acquired on the Orbitrap (*m/z* 400–2000) with a resolution of 60,000. Collision-induced dissociation (CID) fragmentation in the linear ion trap was performed for the five most intense ions. Lipid analysis was replicated three times.

**Statistics.** Liposomal flux assay: Each liposomal flux experiment was performed across at least three independent reconstitution experiments. Each independent measurement was the result of averaging between two and six technical replicates. To assess significance, a two-sided Dunnett's multiple comparisons test was carried out to compare each experimental condition to either channel-free liposomes or wild type-KirBac3.1 proteoliposomes. Significance was determined by a family-wide threshold of 0.05, with the P-value for each comparison adjusted for multiplicity. For each experimental condition, the number of independent (and constitutive technical) replicates and P-values as determined by the Dunnett's multiple comparisons are summarised in Supplementary Tables 3 and 4.

Crystallography: Crystallographic statistics are provided in Supplementary Table 1.

**Reporting summary.** Further information on research design is available in the Nature Research Reporting Summary linked to this article.

## Data availability

Source data are provided with this paper. Atomic coordinates and structure factors generated in this study have been deposited in the Protein Data Bank with accession codes 7N9L and 7N9K. Molecular simulation, fluorimetric assay and electrophysiological data generated in this study are provided as source data and have been deposited on the Figshare server as Source_data_for_Jin_et_al [https://doi.org/10.6084/m9.figshare.c.5752604.v1]. Mass spectrometry lipidomics raw data generated in this study have been deposited on the Figshare server as Lipidomics raw data for KirBac [https://doi.org/10.6084/m9.figshare.17197811.v1]. Source data are provided with this paper.

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

## Acknowledgements

We thank Mike Lawrence, Xiaowen Xiao and Ziyi Meng for advice and assistance. Diffraction data were collected on the MX2 beamline at the Australian Synchrotron, Victoria, Australia. Crystallisation trials were carried out by Janet Newman at the C3 Parkville Facility. This work was funded by Australian Research Training Scholarships (R.J. and S.H.); Medical Research Council Grant MR/V028839/1 (C.V.R. and J.R.B.); Wellcome Trust Grant No. 104633/Z/14/Z (C.V.R. and D.W.) and for the purpose of Open Access, the author has applied a CC BY public copyright licence to any Author Accepted Manuscript version arising from this submission; J.R.B. holds a Royal Society University Research Fellowship; NHMRC Senior Principal Research Fellowship 1116934 (P.M.C.); Australian National Computational Infrastructure, provided through Intersect Australia Ltd. with support from the Australian Government through LIEF grant LE170100032, and the HPC-GPGPU Facility that was established with the assistance of LIEF Grant LE170100200 (B.J.S., S.H. and R.J.); NSW Health infrastructure grant through the Hunter Medical Research Institute (D.L. and P.J.); Victorian State Government Operational Infrastructure Support and Australian Government NHMRC IRIISS.

## Author contributions

B.J.S., R.J. and S.H. contributed to experimental design and conceptual development and performed all molecular simulations. K.A.B contributed to experimental design, protein biochemistry, assay development and crystallography. J.R.B., D.W. and C.V.R. conceived of and performed the mass spectrometry experiments. D.L. and P.J. carried out electrophysiological recordings and analyses. A.W. and P.C. contributed to protein expression. A.P. assisted with analysis and interpretation. P.M.C. contributed to experimental design and interpretation. O.B.C. assisted with analysis. J.M.G. conceived the project, devised and guided the experimental work, contributed to the crystallography and wrote the manuscript.

## Competing interests

The authors declare no competing interests.
