## [Peer Review File · Nature Communications]

Ion currents through Kir potassium channels are gated by anionic lipids.REVIEWER COMMENTS

Reviewer #1 (Remarks to the Author):

Reviewer: Diomedes Logothetis

Jin and colleagues from the Gulbis and Smith labs in Australia present evidence for a novel role for the acyl chains of anionic phospholipids occupying fenestrations on the wall of the pore and interacting through hydrophobic interactions with L124 of the KirBac3.1 channel. The authors employ an impressive array of diverse tools, including simulations (MD and CG), crystallography, Mass Spec, ACMA fluorimetric assay in channels reconstituted in liposomes and single channel activity in channels reconstituted in lipid bilayers. First, they show strong data supporting the role of aliphatic side chains as the channel gates (L124M or Y133I). Then in Figs. 2 and 3, they show both computationally and experimentally that hydrophobic interactions of acyl chains with L124 lead to gating. They conclude the study by showing again both computationally and experimentally that anionic phospholipids PE and PG in particular are tightly associated and retained in the channel. All experiments are well-performed and controlled. The suggestions below aim to further strengthen this very nice study.

Major point

The headgroup interactions are divorced from the acyl chain interactions with the channel gate. Yet, the authors should have all the data needed from their simulations to analyze whether the L124 hydrophobic interactions with the acyl chains that lead to gating events and ion permeation are correlated with specific electrostatic interactions of the phospholipid headgroup with specific channel residues. Coupling the two parts (headgroup and acyl chains) to show how the electrostatic interaction drives the hydrophobic gating event would strengthen the study considerably, as there is no comparable study that has demonstrated such a mechanistic link.

Minor points

- 1) The paper tends to be dense. The authors can improve in communicating the essence of the methodology used and the data plotted so that they minimize the time the non-expert reader will spend trying to understand what they are looking at.
- 2) It was not clear from the figures shown that gating occurred during the course of the simulations and K⁺ ion permeation took place, even though the authors indicate that this happened. A supplemental

figure to highlight this point would be good. The direction of ion flow, how many ions conducted in the course of the simulations and the conditions that led to ion permeation need to be clarified.

3) The existence of fenestrations occupied by lipids has been seen in more than K2P channels (ref. 22 cited by the authors) and up to the present study they have been a curiosity. The authors could expand the list of channels showing lipid occupied fenestrations and given where these fenestrations exist discuss functional consequences the acyl chains could have influencing the function of the channels they are found in.

Reviewer #2 (Remarks to the Author):

This is a very interesting manuscript. I agree that the explicit participation of lipids to fine-tune the opening of the permeation pathway and, thus, gate the channel represents an important and meaningful conceptual shift to the classical view. The data with alky-MTS reagents shows that ion permeation is contingent upon lipid tails being of sufficient length to reach the occlusion. The paper relies on a combination of experimental and computational results and this makes up a fairly compelling story.

Main points:

1) In figure 2 you plot the 2D histogram from "all simulations" and "simulations with ions passing L124". Your mechanistic hypothesis is that the L124 gate is open when there is a contact with lipids (orange in fig 2b) and the gate is open when there isn't a contact. Your figure almost shows this, but not quite. However, by comparing "all simulations" and "simulations with ions passing", it is not completely clear because the L124 side chains must be separated when an ion is passing. So this leaves the question: are the L124 separated because the ion is passing or is the ion passing because the L124 are separated? The real gating question is whether the L124 are separated by the presence of the lipids. To answer this question you should compare two histograms: one made from the snapshots when "lipids are making a contact with L124" and one from the snapshots when the "lipids are not making contact with L124". This comparison will go to the heart of the gating question more directly.

2_ More K⁺ ions passed the Leu124 collar in simulations where all four fenestrations were blocked by a gauche conformation of Tyr57. Can you confirm that the rotameric state of Tyr57 does not affect or distort the structure locally? Looking at the structure it is not clear why the rotamer of Tyr57 blocks access to the lipids. From the point of view of biology, is it possible to envision a mutation that would

bloc the access of lipid chains through the fenestrations? Perhaps I am wrong but a quick look at the structure suggests that substituting a large sidechain at A127, G123, M121 could do this. Could it be tested in MD first and then experimentally?

Minor points

1) Not sure whether the data from Figure 2d comes from unbiased MD or umbrella sampling. Please clarify in the figure caption.

2) A constant electric field is applied along the Z axis of the periodic MD box. In a box of dimensions 8.2 by 8.2 by 8.4 nm, the field 8.4 nm X 50 mV/nm means that a potential of 420 mV was applied across the MD box. However, the potential will be compressed in the aqueous phase to become a potential of 420 mV across the membrane (Roux, Biophys J 2008). This should be stated clearly.

3) Figure 1d could also add the profile of the pore radius (like produced by HOLE for example)

Reviewer #3 (Remarks to the Author):

In this study, the authors employed a combination of structural studies, X-ray crystallography, in silico molecular dynamics, lipid occupancy, electrophysiology, and mass spectrometry to investigate the hypothesis that gating of Kir channel is operated by fatty acyl tails of lipids that infiltrate the conduction pathway via fenestrations in the wall of the pore. It is a very interesting work, with original conclusion. However, there are several concerns, which need to be addressed.

1) The definition of the 'closed' and 'open' state'. The definition of these states needs to be solidified. What are the minimal distances of the pore and the ion diameter of permeating ions.

Line 35 'The narrow conformation is not 'closed' in any physiological, however as K⁺ is able to pass through an opening only marginally wider than its ionic diameter'' What is this precise ionic diameter and is the K⁺ dehydrated or semi-hydrated?

2) 'ion conduction is not gated at the Tyr132 collar'. This claim should be discussed. It is in contradiction with many structural crystallographic data and in silico studies where it is clearly shown that Tyr132 is associated with a gating.

The authors also rely on their investigation on the free energy barrier (potential of mean force PMF), which is 13 kJmol⁻¹ for Leu 124, compared with 6kJmol⁻¹ at Tyr 132. They also claim that these values are not a significant barrier to ion passage. What is the value for which it is a significant barrier?

We understand that even if constriction points exist, they might not be completely occluded, however more details should be given on the pore diameter (do we have the presence of a central hydrophobic zone and what is the size of this pore when Leu residues change conformation?). The Leu residues are flexible, does PMF value vary according to the conformers.

The knowledge of the friction (or diffusion) coefficient of ion motion along the pore axis could also be calculated, and in combination with the knowledge of the PMF values, this would give eventually the conductance of the pore to compare with the experimental data.

3) On Figure Supplementary 2F, the extra density is very difficult to interpret clearly. The authors interpret this density as lipid density and lipid fragments. This density could be attributed to detergent molecules.

4) Figure Sup Figure 3 should provide also the single channel recording for WT at -60mV and -70 mV for comparison.

5) The authors have shown, using MD simulations, that lipids engaging Leu124 side chain is correlated to the opening of the Leu124 collar. MD simulation with coarse grain, is expected to show large movement during the gating. Movement of lateral chain of Leu124 and lipidic chain are sufficiently large to be seen by this method. Can the authors provide a movie?

6) PIP2 is a necessary co-factor for mammalian Kir gating, while it has been reported to inhibit prokaryotic Kir activity. PIP2 was mentioned in the present work. It would be useful for the authors to elaborate and discuss differences between prokaryotic and eukaryotic Kir channels with respect to gating by PIP2.

Minor comment:

C72S should be C71S

REVIEWER COMMENTS

We are conscious of the time and effort that goes into reviewing manuscripts and thank all reviewers for their measured and thoughtful commentary, and for their constructive suggestions that strengthen the manuscript.

Reviewer #1 (Remarks to the Author):

Reviewer: Diomedes Logothetis

Jin and colleagues from the Gulbis and Smith labs in Australia present evidence for a novel role for the acyl chains of anionic phospholipids occupying fenestrations on the wall of the pore and interacting through hydrophobic interactions with L124 of the KirBac3.1 channel. The authors employ an impressive array of diverse tools, including simulations (MD and CG), crystallography, Mass Spec, ACMA fluorimetric assay in channels reconstituted in liposomes and single channel activity in channels reconstituted in lipid bilayers. First, they show strong data supporting the role of aliphatic side chains as the channel gates (L124M or Y133I). Then in Figs. 2 and 3, they show both computationally and experimentally that hydrophobic interactions of acyl chains with L124 lead to gating. They conclude the study by showing again both computationally and experimentally that anionic phospholipids PE and PG in particular are tightly associated and retained in the channel. All experiments are well-performed and controlled. The suggestions below aim to further strengthen this very nice study.

We appreciate these comments and thank the reviewer.

Major point

The headgroup interactions are divorced from the acyl chain interactions with the channel gate. Yet, the authors should have all the data needed from their simulations to analyze whether the L124 hydrophobic interactions with the acyl chains that lead to gating events and ion permeation are correlated with specific electrostatic interactions of the phospholipid headgroup with specific channel residues. Coupling the two parts (headgroup and acyl chains) to show how the electrostatic interaction drives the hydrophobic gating event would strengthen the study considerably, as there is no comparable study that has demonstrated such a mechanistic link.

Homology to eukaryotic Kir channels suggests residues involved in phospholipid head binding via the phosphates include the ¹³⁷RP¹³⁸ motif on the interdomain linker between inner helix and intracellular domain, and Trp46, which is located at the junction of the slide helix and outer helix. Our simulations implicate both of these and also His37 (from the slide helix of the adjacent subunit). Not included in our simulation model is Thr136, which is well placed to hydrogen-bond to any phospholipid head group.

Bearing in mind that some of the phosphate interactions with PI(4,5)P2 are water-mediated (e.g. see Kir2.2; PDB 3SPI), we enumerated the proximity (within 5 Å) of a phosphate oxygen of each lipid to the (RP, H, W) cluster of side chains. 95.5 % of the lipids with tails occupying fenestrations had head groups falling within this 5 Å proximity limit, indicating that fenestration-occupying lipids almost invariably have tightly bound head groups, connecting gating at the fenestrations to changes at the lipid binding site.

Figure: an isosurface of the fenestration-bound lipids from frames in which the K⁺ ion is found transiting the Leu124 collar demonstrates the relationship of lipid to a canonical lipid-binding site; the isosurface was calculated from 292 lipid molecules (85 events) at a mass density contour level of 0.15. The isosurface is shown

as teal mesh, and sphere representations of the His, Arg and Trp are coloured on an otherwise grey molecular surface.

While our MD simulations were carried out on the transmembrane pore, the head group moieties of PS and PG are large enough to make additional contacts to the proximal intracellular assembly (as PI(4,5)P2 does in Kir channels), through electrostatic interactions with some of many polar residues in the vicinity. Coarse-grained simulations in Figure 4 show PS and PG headgroups condensed near the channel (relative to PC), suggesting that their position is restrained by additional electrostatic interactions to the linker and/or cytosolic domain facilitated by the anionic lipids.

Thus, lipids present in the fenestrations *and* engaging Leu124 make multiple electrostatic contacts to both pore and intracellular elements of the protein, in the vicinity of the tetramer interfaces. A nexus of contacts by the head group across the domain boundary directly couples interdomain motions of the Kir channel (Clarke et al 2010) to the movement of lipid tails engaging Leu124, which operate the gate.

We have added a paragraph to the text on page 7: “For the purpose of clearly defining any relationship between the fenestration-bound lipid tails and head group binding, we enumerated the proximity of the phosphate of the lipids to the lipid binding site using the all-atom MD data. This revealed that even with neutral PC, 95.5 % of the lipids occupying fenestrations had head groups falling within 5 Å of the Arg137, Trp46 and His37 pocket, indicating that lipids occupying fenestrations almost invariably have tightly bound head groups. An isosurface of the fenestration-bound lipids from frames in which the K⁺ ion is found transiting the Leu124 collar demonstrates the relationship of lipid to a canonical lipid-binding site (Fig. 4c). The condensation of PS and PG headgroups revealed in the coarse-grained simulations suggest additional electrostatic contacts facilitated by anionic lipids in pore and intracellular assembly are critical in attracting and confining them. The nexus of electrostatic contacts to both pore and intracellular elements of the protein in the vicinity of the tetramer interfaces is a prerequisite for direct coupling of interdomain motions of the Kir channel⁴ to the movement of lipid tails engaging Leu124”.

In the discussion, we amended the paragraph starting at line 264 ” We have established that the aliphatic termini of lipids ...” to incorporate this mechanistic link.

Minor points

1) The paper tends to be dense. The authors can improve in communicating the essence of the methodology used and the data plotted so that they minimize the time the non-expert reader will spend trying to understand what they are looking at.

We have incorporated small changes throughout to simplify.

2) It was not clear from the figures shown that gating occurred during the course of the simulations and K⁺ ion permeation took place, even though the authors indicate that this happened. A supplemental figure to highlight this point would be good. The direction of ion flow, how many ions conducted in the course of the simulations and the conditions that led to ion permeation need to be clarified.

Ion flow is inward, from the upper vestibule toward the cytosol.

In a total of 14 μs of unrestrained simulations (70 x 200 nsec) on the wild type channel, a total of 85 conduction events occurred. Ion permeation coincides with high fenestration occupancy and a corresponding increase in the breadth of the ion conduction pathway at Leu124 sufficient to accommodate a partially hydrated K⁺. Comparison of the permeation data to Fig. 3a (all simulations) shows the key criterion for permeation is an average cross-sectional aperture in the 40-50 Å² range, and that this corresponds to a generally higher number of lipids engaging Leu124 (a shift from 2 to 4 occupying lipids), although even a single occupying lipid is (very rarely) sufficient.

We have amended the penultimate paragraph of the Discussion to start “Moreover, we have demonstrated using MD simulations that the key criterion for permeation is widening of the Leu124 collar, and that this is correlated to the number of lipids engaging Leu124 side chains.”

Suppl. Fig. 5c. Analysis of lipid occupancy and aperture area for permeation events. The left panel shows the distribution of lipids engaging Leu124 via fenestrations, while the right panel shows the distribution of aperture cross-sectional area and correlates it to the number of lipid-occupied fenestrations (mean±SD).

3) The existence of fenestrations occupied by lipids has been seen in more than K2P channels (ref. 22 cited by the authors) and up to the present study they have been a curiosity. The authors could expand the list of channels showing lipid occupied fenestrations and given where these fenestrations exist discuss functional consequences the acyl chains could have influencing the function of the channels they are found in.

We concur that specific lipid binding in fenestrations has been observed in a diverse range of ion channels, including, but not limited to, K2P1 (PDB 3UKM) and TRIC, a trimeric intracellular cation channel (PDBs 5H36, 5H35). Experimental data (photo-active lipids) have implicated fenestration lipids in influencing channel activity in non-selective TRPC3 channels and MD simulations of NaVAb. MD simulations of MthK, a Ca²⁺-activated K⁺ channel, have shown lipid moving readily into K⁺ channel fenestrations *in silico*.

Tightly bound anionic lipids appear likely to be involved in gating all K⁺-selective and other ion channels, as their effects have been noted on multiple K⁺ channels, but one would expect the nuance, or mechanistic detail, to differ in accord with each ion channel family. The knowledge that K⁺ can pass through very small openings by partial dehydration obviates a requirement for largescale conformational changes during K⁺ channel gating. This, and the findings presented here, carry implications for all K⁺ channels. Even so, we have confined our remarks mainly to Kir channels, where we have strong evidence to support them.

We have shifted and amended a sentence in the conclusions to the end of the discussion, including additional references therein: “Our findings sit within an emerging context of a general sensitivity of ion channels, including all major classes of K⁺ channels, to phosphoinositides and anionic lipids in general. Analogous fenestrations have been reported in diverse ion channel families, structures including but not limited to TREK-2, K2P1 and TRIC, and experimental data implicating fenestration lipids in TRPC3 activity, suggesting the concept of specifically bound phospholipids as allosteric gating elements may be generally applicable”

Reviewer #2 (Remarks to the Author):

This is a very interesting manuscript. I agree that the explicit participation of lipids to fine-tune the opening of the permeation pathway and, thus, gate the channel represents an important and meaningful conceptual shift to the classical view. The data with alky-MTS reagents shows that ion permeation is contingent upon lipid tails being of sufficient length to reach the occlusion. The paper relies on a combination of experimental and computational results and this makes up a fairly compelling story.

We appreciate the positive outlook of this reviewer and their recognition of an important conceptual shift in the field triggered by this study.

Main points:

1) In figure 2 you plot the 2D histogram from "all simulations" and "simulations with ions passing L124". Your mechanistic hypothesis is that the L124 gate is open when there is a contact with lipids (orange in fig 2b) and the gate is open when there isn't a contact. Your figure almost shows this, but not quite. However, by comparing "all simulations" and "simulations with ions passing", it is not completely clear because the L124 side chains must be separated when an ion is passing. So this leaves the question: are the L124 separated because the ion is passing or is the ion passing because the L124 are separated? The real gating question is whether the L124 are separated by the presence of the lipids. To answer this question you should compare two histograms: one made from the snapshots when "lipids are making a contact with L124" and one from the snapshots when the "lipids are not making contact with L124". This comparison will go to the heart of the gating question more directly.

This comment is most helpful. The data presented in Fig 4d shows the aperture at the Leu124 collar is sufficiently wide to permit a K^+ ion to pass significantly longer ($\sim 35,000$ frames/bin) than those with the K^+ ion passing the collar (~ 250 frames/bin), thus answering the reviewer's question – the ion passes because the collar is open. To assist the reader, we have provided a one-minute movie (a 150 nsec simulation) of an ion being conducted through KirBac3.1. Concomitant orthogonal views show lipid tails (pink) moving in and out of the fenestrations to engage Leu124 (yellow) throughout. The Leu124 collar widens and narrows during the simulation, regardless of whether an ion is close by. When an ion is in the vicinity, it passes only when the collar is sufficiently wide. We have included this as a supplementary movie (Suppl. Movie 1) and referenced it on line 125.

We amended the text, starting line 119, to "Ions slipped past the Leu124 collar when the mean distance across the opening is approximately 5 Å in diameter (van der Waals' accessible surface) (Fig. 2d, Supplementary Fig. 5a-b), or 40-50 Å² in cross-sectional area Supplementary Fig. 5c-d. Each K^+ retained a coordination shell of four to five water molecules as it passed through (Fig. 2e). Figure 2d shows a probability density of approximately 35,000 frames/bin for an aperture at the Leu124 collar of sufficient size for conduction; the comparable probability density when an ion is simultaneously passing is far less, at approximately 250 frames/bin. This resolves any issue of whether the ion pushing past is the cause of collar opening; the Leu124 collar opens as a consequence of lipids engaging Leu124 residues (Supplementary Movie 1)."

2_ More K^+ ions passed the Leu124 collar in simulations where all four fenestrations were blocked by a gauche conformation of Tyr57. Can you confirm that the rotameric state of Tyr57 does not affect or distort the structure locally? Looking at the structure it is not clear why the rotamer of Tyr57 blocks access to the lipids. From the point of view of biology, is it possible to envision a mutation that would block the access of lipid chains through the fenestrations? Perhaps I am wrong but a quick look at the structure suggests that substituting a large sidechain at A127, G123, M121 could do this. Could it be tested in MD first and then experimentally?

We appreciate these constructive suggestions and can confirm that we could find no evidence of local distortion in our MD simulations. Restraining the Tyr57 side chain in a *gauche* rotamer was the least intrusive, most effective, means we could find of preventing lipid from entering far enough to engage with Leu124. In the *gauche* rotamer of Tyr57, the only likely clash is with the flexible side chain of Met121 in one of its adopted rotamers (noting that in structures of KirBac3.1, the Met121 side chain is typically poorly ordered and modelled with alternate rotamer conformations).

Figure: Superposition of Tyr57 *trans* rotamer (crystal structure) and Tyr57 *gauche* (MD simulation), also showing Met121 adopting different rotamers; in the crystal structure both are observed and modelled as alternate side chain conformers but in the Tyr57 *gauche* rotamer only one is possible.

In MD simulations of the wild-type pore, Tyr57 resides in the *gauche* conformation in ~11% of the frames (Suppl. Fig. 6a), indicating the compatibility of the *gauche* rotamer with the local structure. Moreover, in structures of eukaryotic Kir channels (example PDB codes: 3SPI, 6XIT, 7MJO) the conserved aromatic residue at this position is most often seen adopting the *gauche*

rotamer.

Additional histograms presented below clarify that when a lipid tail is occupying a fenestration, the Tyr57 rotamer can only be *trans* (i.e. lipid entry is incompatible with a *gauche* Tyr57 rotamer). Conversely, if the Tyr57 side chain is restricted to *gauche*, only 1% of simulations show a lipid in that fenestration.

This figure, now added to Suppl. Fig. 6, as panel (e), shows how the view from the fenestration entrance (transparent molecular surface) differs when Tyr57 adopts a *trans* rotamer (left; crystal structure) or a *gauche* one (right; Y57-gauche MD simulation model).

Tyr57 is represented by yellow (carbon) and red (oxygen) spheres, and the lipid/detergent alkyl chain from the transposed crystal structure by orange spheres. The figure shows the alkyl chain collides with the *gauche* rotamer of Tyr57.

We have revised the text starting at line 140 to: “...we conservatively occluded the

fenestrations by altering a neighbouring tyrosine (Tyr57) from the *trans* rotamer observed in the crystal structure (χ_1 (N-C α -C β -C γ) = 180°) and constraining it in a *gauche* (χ_1 = -60°) conformation, a naturally occurring but low frequency rotamer in KirBac3.1 simulations (Supplementary Fig. 6a).”

We considered several possibilities for blocking lipid entry to fenestrations, to identify a means of occlusion with minimal structure perturbation. Local disorder of Met121 (see above) makes it untenable, whereas mutating A127 to a larger side chain results in short (< vdW) contacts to other inner and outer helix residues, potentially producing local structural distortions that would complicate interpretation of the results. Similarly, altering the hydrogen-bonding at Gly123, adjacent to Leu124 in the inner helix, would likely hamper the critical Leu124 movement (*i.e.* the dual effects would complicate interpretation).

As a minimally disruptive experimental means of achieving fenestration block, we used a hexyl substituent to mimic the natural acyl tail. As the fenestration occupancy proved low, rather than zero, we performed C119-hexyl simulations for direct comparison. Suppl. Fig. 10d-f shows the non-zero lipid occupancy in C119-hexyl simulations; one Leu is engaged ~20% of the time, and two are engaged ~5% of the time (with correspondingly rare conduction events). This additionally provides a plausible explanation for why, in C119-hexyl derivatised channels, it was far harder to locate active channels, and activity was largely limited to a sub-conductance level (Fig. 3), correlating leucine engagement to subconductance/conductance status.

Minor points

1) Not sure whether the data from Figure 2d comes from unbiased MD or umbrella sampling. Please clarify in the figure caption.

The data is from unbiased, unrestrained, MD simulations.

We have amended the sentence in the figure caption to: “Hexagonal bin plots of D1 against D2 (aperture diagonals) for structures extracted from unbiased MD simulations”.

2) A constant electric field is applied along the Z axis of the periodic MD box. In a box of dimensions 8.2 by 8.2 by 8.4 nm, the field 8.4 nm X 50 mV/nm means that a potential of 420 mV was applied across the MD box. However, the potential will be compressed in the aqueous phase to become a potential of 420 mV across the membrane (Roux, Biophys J 2008). This should be stated clearly.

In the methods, we have included the statement “All unrestrained simulations were carried out with a static external electric field with a strength of 0.05 V nm⁻¹ applied along the membrane normal direction; a potential across the simulation box of approximately 420 mV operates across the membrane⁶”.

3) Figure 1d could also add the profile of the pore radius (like produced by HOLE for example).

We have included an additional panel as Fig. 1b – using HOLE to calculate a profile of the pore radius.

Reviewer #3 (Remarks to the Author):

In this study, the authors employed a combination of structural studies, X-ray crystallography, in silico molecular dynamics, lipid occupancy, electrophysiology, and mass spectrometry to investigate the hypothesis that gating of Kir channel is operated by fatty acyl tails of lipids that infiltrate the conduction pathway via fenestrations in the wall of the pore. It is a very interesting work, with original conclusion.

We appreciate this affirmation and recognition of the originality of the conclusions presented in this study.

However, there are several concerns, which need to be addressed.

1) The definition of the 'closed' and 'open' state'. The definition of these states needs to be solidified.

We agree that this is a thorny issue. In structural biology, open and closed terminology have been used to refer to the width of the opening to the ion conduction pathway located at the inner helix bundle, where open refers to an aperture sufficiently wide to accommodate a fully hydrated K^+ ion and closed is too narrow to accommodate it. By contrast, in electrophysiology, open and closed are common parlance for conducting (activated) and non-conducting (deactivated or resting) states, respectively. The two concepts have been erroneously treated as interchangeable, but one refers to structure and the other to function. Structurally, conformation appears to be conserved within families (i.e., sequence-driven) and we have found no unequivocal/compelling evidence that wide and narrow pore conformations respectively equate to 'open' and 'closed' gating states.

We have amended a few words in the text to restrict the use of 'open and closed' terminology to functional measurements, with the exception that the Leu124 cluster may descriptively be termed open collar – as this is the only region of the transmembrane pore that significantly alters in girth.

The main change is in the introductory paragraph, now: "Until recently, it was thought that K^+ channels must adopt a wide pore conformation to enable ion permeation. However, K^+ can be conducted through a constriction at the intracellular entrance only marginally wider than its ionic diameter, due to transient depletion and replenishment of its hydration shell. The capacity of K^+ ions to move through a constricted inner helix bundle has re-opened the question of gating, or how Kir channels achieve controlled switching between conducting and non-conducting physiological states."

What are the minimal distances of the pore and the ion diameter of permeating ions.

Line 35 'The narrow conformation is not 'closed' in any physiological, however as K^+ is able to pass through an opening only marginally wider than its ionic diameter"

What is this precise ionic diameter and is the K^+ dehydrated or semi-hydrated?

Pauling's calculated *ionic* radius of K^+ is 1.33 Å. Shannon gives the *ionic* radius of K^+ as 1.38 Å, based on crystallographic data (Shannon, R. D. *Acta Crystallogr Sect Cryst Phys Diffr Theor Gen Crystallogr* 32, 751–767, 1976). The time-averaged *hydrated* radius of K^+ is cited as ~3.4 Å (Barthel, J. & Jaenicke, R. B. E. *Conway: Ionic Hydration in Chemistry and Biophysics. —Vol. 12 Studies in Physical and Theoretical Chemistry. Elsevier, Amsterdam and New York 1981; Dove, P. M. & Nix, C. J. *Geochim Cosmochim Acta* 61, 3329–3340, 1997). The first paragraph of the Introduction now explicitly notes the ionic radius of K^+ . In the simulations, most potassium ions pass the Leu124 collar as partially hydrated species with 4 - 5 coordinating water molecules (Fig. 2e).*

2) 'ion conduction is not gated at the Tyr132 collar '. This claim should be discussed. It is in contradiction with many structural crystallographic data and in silico studies where it is clearly shown that Tyr132 is associated with a gating.

We referred to our recently published study (*Black et al, Nature Commun. 2020*), that showed definitively that the constriction at Tyr132 is not a gate to K⁺ permeation. In brief, we covalently constrained the pore of KirBac3.1 at the constricted inner helix bundle (Tyr132), locking all four inner helices such that the conduction pathway at the inner helix bundle was unable to widen sufficiently to accommodate hydrated K⁺; importantly, the crosslinks did not impair function.

To verify that the disulfide linkages were truly effective at restricting opening, we used the natural Kir channel blocker spermine, which enters the pore from the intracellular side, to gauge the size of the opening (at ~ 4 Å, the cross-sectional diameter of spermine is intermediate between naked and fully hydrated K⁺). While spermine blocked wild-type as expected, it was unable to block a disulfide-linked mutant; indicating the disulfide-linked intracellular opening was sufficiently large to conduct potassium ions but too small to accommodate spermine. Implicitly, the opening is too small to accommodate fully hydrated K⁺. The comparison indicated that the permeating K⁺ species in Kir channels is only partially hydrated and that the narrow opening at Tyr132 is insufficient to gate ion flow.

The reference to our earlier work is included in the text.

The authors also rely on their investigation on the free energy barrier (potential of mean force PMF), which is 13 kJmol⁻¹ for Leu 124, compared with 6kJmol⁻¹ at Tyr 132. They also claim that these values are not a significant barrier to ion passage. What is the value for which it is a significant barrier?

To clarify, 6 kJ mol⁻¹ is consistent with a diffusion-limited process. A review on the mechanism of permeation in the selectivity filter from the de Groot and Kopec groups (*J Mol Biol* **433**, 167002, 2021) notes the barrier to permeation in the selectivity filter has been estimated to be 2-4 kcal mol⁻¹ (~8 - 17 kJ mol⁻¹) in a variety of potassium channels, and therefore the process is essentially diffusion limited. Another study (*J Gen Physiology* *151*, 1231–1246, 2019) that calculated a PMF through the related channel Kir3.2 over the entire width of the membrane found a PMF faced by K⁺ within an intact conducting selectivity filter of ~15 kJ mol⁻¹. Thus, in the context of ion conduction, any barrier less than ~17 kJ mol⁻¹ is not significant.

We have referenced the de Groot review at the appropriate juncture.

We understand that even if constriction points exist, they might not be completely occluded, however more details should be given on the pore diameter (do we have the presence of a central hydrophobic zone and what is the size of this pore when Leu residues change conformation?). The Leu residues are flexible, does PMF value vary according to the conformers.

We appreciate these thoughtful points. The pore cavity of KirBac3.1 between the polar residues Thr96 (selectivity filter) and Tyr132 is indeed lined with hydrophobic side chains. In crystal structures of native KirBac3.1, the Leu124 side chains invariably adopt the same rotamer (chi1 180, chi2 180), where mutual steric attraction of the leucine side chains results in a steric plug dividing the conduction pathway into upper and lower cavities. The accessible pore diameter at the Leu124 collar in the crystal structures (~2.5 Å) is less than the ionic diameter of K⁺ (which is just shy of 2.8 Å). For comparison, we show an internal surface representation of the KirBac3.1 pore from the crystal structure, calculated in HOLE, and a conducting simulation structure where Leu124 is engaged by lipid; included as Supplementary Fig. 5d.

Comparative internal cavity profiles for the crystal structure (where Leu124 occludes the conduction path) and a representative MD conducting structure (where Leu124 is engaged by lipid).

The panels pictorially represent the conduction path dimensions output from HOLE (solid surface). The key difference between the structures is the release (expansion) of the constriction in the conduction pathway at the Leu124 collar.

As the reviewer alludes, the PMF free energy is based on all possible states along the reaction coordinate, thereby accounting for all possible Leu124 conformers. Thus, in a membrane, when lipids draw the Leu124 side chains away from the conduction pathway, the local PMF is determined from all possible Leu124 conformations.

Altering the rotamer of the sidechains of Tyr57 allowed us to prevent lipid engaging the Leu124 side chains, thereby defining a reaction coordinate that excludes states in which the leucine sidechains are withdrawn from the conduction pathway (*i.e.* the conduction pathway is always occluded, similarly to the crystal structure). This enabled the estimation of the PMF along a reaction coordinate that excluded the involvement of the lipid tails on the energy profile, resulting in a substantially higher PMF of 41 kJ mol⁻¹ and clarifying the importance of the lipid-leucine interaction.

The knowledge of the friction (or diffusion) coefficient of ion motion along the pore axis could also be calculated, and in combination with the knowledge of the PMF values, this would give eventually the conductance of the pore to compare with the experimental data.

The frictional coefficient cannot be calculated from our simulation data – this would require the PMF calculated over the entire transmembrane region, including the selectivity filter; the calculations presented here are focused on the region proximal to the Leu124 collar.

3) On Figure Supplementary 2F, the extra density is very difficult to interpret clearly. The authors interpret this density as lipid density and lipid fragments. This density could be attributed to detergent molecules.

An excellent observation – thank you. While we know from the native mass spectrometry (Fig. 4) that tightly bound lipids carry through the purification process, the protein was crystallised from detergent-extracted material. As it is not possible to distinguish whether the long alkyl chains are lipid or detergent fragments, we have revised the figure caption to “several fragments of lipid or detergent.”.

4) Figure Sup Figure 3 should provide also the single channel recording for WT at -60mV and -70 mV for comparison.

For the liposomal fluorimetric assay, where we compare overall activity of the mutants to wild type, we made liposomes incorporating natural *E. coli* lipid extracts. These experiments showed that the L124M mutant was more active than WT.

However, for the single channel recordings we sought more restricted lipid conditions (no lipid extracts) to observe explicit effects of lipomimetics and point mutations without confounding factors (such as bulk membrane lipid contributions to Leu124 gating). We made use of internal controls between samples. In these artificial bilayers, wild type channels are inactive, but L124M is active, allowing us to verify that L124M has no requirement for lipid (consistent with ablation of the Leu124 ‘gate’ and the negligible PMF at both Leu124 and Tyr132 constrictions). The L124M recordings in Suppl. Fig. 3 provide an internal control for the C119-alkyl recordings in Fig. 3, which were carried out under the same conditions, allowing us to directly compare the effect of the various alkyl derivatives

on function without having to make allowances for the possibility of gate operation by bulk membrane lipids.

We have, however, previously published wild type traces for wild type using GUVs made from PC extracts (Clarke et al, 2010; referenced) which give a representative functional profile under less stringent lipidic conditions.

5) The authors have shown, using MD simulations, that lipids engaging Leu124 side chain is correlated to the opening of the Leu124 collar. MD simulation with coarse grain, is expected to show large movement during the gating. Movement of lateral chain of Leu124 and lipidic chain are sufficiently large to be seen by this method. Can the authors provide a movie?

Yes. A movie showing changes at the fenestrations and Leu124 collar during conduction has been provided as Suppl. Movie 1.

6) PIP2 is a necessary co-factor for mammalian Kir gating, while it has been reported to inhibit prokaryotic Kir activity. PIP2 was mentioned in the present work. It would be useful for the authors to elaborate and discuss differences between prokaryotic and eukaryotic Kir channels with respect to gating by PIP2.

The reviewer may be referring to a paper from the Nichols lab showing inhibition of KirBac1.1 by PI(4,5)P2 (Cheng et al, *J Gen Physiol* 133, 295–305, 2009); this lipid is not found in the KirBac1.1 host organism *Burkholderia pseudomallei* and we are unaware of any study that extends this finding to other prokaryotic Kir channels. While PI(4,5)P2, as an anionic lipid, can bind prokaryotic Kir channels, its effect (e.g. activation, inhibition) and any biological significance is unclear, for reasons given below.

While prokaryotic and eukaryotic Kir channels share a canonical lipid binding site that accommodates a range of lipids, most eukaryotic Kir channels appear to be activated only by PI(4,5)P2, i.e. other lipids can bind but do not confer function. Phosphoinositides are primarily found in eukaryotes; while there are none in *E. coli* (Botero et al *Sci Adv* 5, eaat4872, 2019), myo-inositides have been found in some groups of bacteria and archaea (Koga, Y. & Morii, *Biosci Biotechnology Biochem* 69, 2019–2034, 2014). In bacteria, phosphatidylglycerols represent the main anionic phospholipid class but in *Magnetospirillum magnetotacticum*, from which KirBac3.1 is derived, phosphatidylserine is also present. Thus, PS and PG lipids are more relevant to KirBac3.1 function than PI(4,5)P2.

Prokaryotic and eukaryotic Kir channels are highly homologous structurally, however a flexible three-residue inclusion of basic residues in the interdomain linker of eukaryotic channels (see introduction) increases the separation between the pore and intracellular domains. Some phospholipids (e.g. PI(4,5)P2), and even detergents, (Nishida et al. *EMBO J* 26, 4005–4015, 2007) bound to eukaryotic Kir channels draw these domains together, conferring a spatial relationship between the pore and intracellular domains that is the same as in prokaryotic Kir channels. This change is necessary, but insufficient, to confer activity.

Under current circumstances, where the mechanism of PIP2 gating has not been definitively elucidated, we feel it would be premature to initiate a dialogue comparing the two.

Minor comment:

C72S should be C71S

Apologies. This has been remedied.

REVIEWERS' COMMENTS

Reviewer #1 (Remarks to the Author):

The authors have given thoughtful responses and addressed all my points, major and minor. I am satisfied with their answers and support publication.

Reviewer #2 (Remarks to the Author):

The authors have answered convincingly all my remarks. In figures 2d comparing the panels for all simulations with "ion passing" gets at the correct argument about the lipid-gating mechanism but is a little circumvoluted. The Supp Movie is great. Ultimately, the most convincing evidence of the lipid-gating mechanism is actually provided by figure 3a showing that the aperture area varies with the number of lipids engaging with Leu124 side chains. These this emphasized in the text (lines 136-139). So I think this point is very clear and I am satisfied. This is a nice manuscript making an important mechanistic point in a compelling fashion.

1) A minor point about Figure 2. From a craftsmanship point of view, the panels in Figure 2d emphasizing the actual frames per bins are a little unusual. It is more common to focus on the normalized histograms, but I can live with this. However, it would be helpful to plot different levels using rainbow colors rather than different intensities of the same purple in Figure 2d. My recommendation would be to replot with different colors for clarity.

2) Regarding the "biased" versus "unbiased" simulations, the text is perhaps somewhat confusing. Commonly, unbiased MD refers to equilibrium simulations with no artificial restraints. Often, "biased" MD refers to simulations with artificial restraints such as targeted MD, steered MD, umbrella sampling, metadynamics, etc. Not such artificial restraints were used here. However, an external electric field was introduced to realistically simulate the transmembrane potential. So, strictly speaking those simulations are perhaps "unbiased but most clearly "nonequilibrium simulations". The text should be corrected for the sake of clarity.

3) Some x-axis labels are overlapping and obscured in Supplementary figure 10 b and d.

Reviewer #3 (Remarks to the Author):

The authors have responded to each of my comments. The answers are clear, detailed, well documented , really interesting and convincing.

Ion currents through Kir potassium channels are gated by anionic lipids.
Jin et al.

We thank all three reviewers for reading through critically a second time and for their constructive and favourable commentary. It has helped us to distil our arguments and present the main points more clearly.

REVIEWERS' COMMENTS

Reviewer #1 (Remarks to the Author):

The authors have given thoughtful responses and addressed all my points, major and minor. I am satisfied with their answers and support publication.

Reviewer #2 (Remarks to the Author):

The authors have answered convincingly all my remarks. In figures 2d comparing the panels for all simulations with "ion passing" gets at the correct argument about the lipid-gating mechanism but is a little circumvoluted. The Supp Movie is great. Ultimately, the most convincing evidence of the lipid-gating mechanism is actually provided by figure 3a showing that the aperture area varies with the number of lipids engaging with Leu124 side chains. These this emphasized in the text (lines 136-139). So I think this point is very clear and I am satisfied. This is a nice manuscript making an important mechanistic point in a compelling fashion.

1) A minor point about Figure 2. From a craftsmanship point of view, the panels in Figure 2d emphasizing the actual frames per bins are a little unusual. It is more common to focus on the normalized histograms, but I can live with this. However, it would be helpful to plot different levels using rainbow colors rather than different intensities of the same purple in Figure 2d. My recommendation would be to replot with different colors for clarity.

To clarify, we used hexbin plots to illustrate changes at the Leu124 collar as we felt they would be the most intuitive to the reader. Regarding the colour scheme, we have tested replotting in a rainbow scheme as requested, but in this instance found it distracting to the eye. Also, it is not in keeping with present editorial policy i.e. "Rainbow gradients can be problematic; use alternative gradients if possible".

2) Regarding the "biased" versus "unbiased" simulations, the text is perhaps somewhat confusing. Commonly, unbiased MD refers to equilibrium simulations with no artificial restraints. Often, "biased" MD refers to simulations with artificial restraints such as targeted MD, steered MD, umbrella sampling, metadynamics, etc. Not such artificial restraints were used here. However, an external electric field was introduced to realistically simulate the transmembrane potential. So, strictly speaking those simulations are perhaps

"unbiased but most clearly "nonequilibrium simulations". The text should be corrected for the sake of clarity.

To clarify this point, the first sentence of the caption for Fig. 1d has been re-amended to "(d) Hexagonal bin plots of D1 against D2 (aperture diagonals) for structures extracted from unbiased non-equilibrium MD simulations."

3) Some x-axis labels are overlapping and obscured in Supplementary figure 10 b and d.

Many thanks. These have been remedied.

Reviewer #3 (Remarks to the Author):

The authors have responded to each of my comments. The answers are clear, detailed, well documented , really interesting and convincing.

** See Nature Research's author and referees' website at www.nature.com/authors for information about policies, services and author benefits